# Space Weather Study through Analysis of Solar Radio Bursts detected by a Single Station CALLISTO Spectrometer

Theogene Ndacyayisenga[1], Ange Cyanthia Umuhire[1], Jean Uwamahoro[2], and Christian Monstein[3]

[1]University of Rwanda, College of Science and Technology, P.O. Box 3900, Kigali – Rwanda.
[2]University of Rwanda, College of Education, P.O. BOX 55, Rwamagana – Rwanda.
[3]Istituto Ricerche Solari (IRSOL), Università della Svizzera italiana (USI), CH-6605 Locarno-Monti, Switzerland.

**Correspondence:** Theogene Ndacyayisenga (ndacyatheogene@gmail.com)

**Abstract.** This article summarizes the results of an analysis of solar radio bursts (SRB) detected by the Compound Astronomical Low-cost Low-frequency Instrument for Spectroscopy and Transportable Observatory (CALLISTO) spectrometer hosted by the University of Rwanda. The data analyzed were detected during the first year (2014 – 2015) of the instrument operation. Using quick plots provided by the e-CALLISTO website, a total of 201 intense and well-separated solar radio bursts detected by the CALLISTO station located in Rwanda, is found consisting of 4 type II, 175 type III, and 22 type IV radio bursts. It is found that all analyzed type II and $\sim 37$ % of type III bursts are associated with impulsive solar flares while the minority ($\sim 13$ %) of type IV radio bursts are associated with solar flares. Furthermore, all type II radio bursts are associated with CMEs, $\sim 44$ % of type III bursts are associated with CMEs, and the majority ($\sim 82$ %) of type IV bursts were accompanied by CMEs. With aid of the Atmospheric Imaging Assembly (AIA) images onboard the Solar Dynamics Observatory (SDO), the location of open magnetic field lines of non-flare associated type III radio bursts are shown. The same images are used to show the magnetic loops in the solar corona for type IV radio bursts observed in the absence of solar flares and/or CMEs. Findings from this study indicate that analysis of SRBs that are observed from the ground can provide a significant contribution to the early diagnosis of solar transients phenomena such as solar flare and CMEs which are major drivers of potential space weather hazards.

## 1 Introduction

The solar space weather events like Coronal Mass Ejections (CMEs) and solar flares are usually accompanied by solar radio bursts (SRBs), which can be used for a low-cost real-time space weather monitoring (Lobzin et al., 2009). SRBs are classified into five types based on their morphologies and drift rates (Wild, 1950). From the meter to decimeter range, characteristic burst signatures correspond to well-identified physical processes, such as shock waves (type II bursts, Nelson and Melrose, 1985; Cairns et al., 2003; Ganse et al., 2012), electron beams streaming along open magnetic field lines (type III bursts, Lin et al., 1981, 1986), or electron populations trapped in eruptive flux ropes and post-flare loops (type IV bursts, Nindos et al., 2008).

Type II radio bursts are the bright radio emissions often associated with CMEs and characterized by a slow frequency drift rate ($\leq -1\ MHz\ s^{-1}$) (McLean and Labrum, 1985; Nelson and Melrose, 1985). They are excited by magneto-hydrodynamics (MHD) shocks in the solar atmosphere (Nelson and Melrose, 1985; Cliver et al., 1999; Nindos et al., 2008, 2011; Vršnak and Cliver, 2008). MHD shocks are driven by both flares and CMEs in the solar atmosphere (Nindos et al., 2008). The type

II radio emissions are generated at the fundamental and second harmonic of the local plasma frequency. Both fundamental and harmonic band can be present, and sometimes each band is split into a higher and a lower frequency lane with a relative frequency splits span in the range $\Delta f/f = 0.05 - 0.6$ (Vršnak et al., 2001, 2002, 2004). Type II radio bursts that are observed in meter wavelength and associated with CMEs can serve as a proxy in estimating the CME kinematics close to the Sun (Shanmugaraju et al., 2017; Kumari et al., 2017a, b). The type III radio bursts are the intense, frequently observed, and fast drifting bursts from high to low frequencies in the dynamic spectra. These bursts usually come from active regions (Saint-Hilaire et al., 2013) and they are generated by electrons propagating along open magnetic field lines and trigger the plasma oscillations (also known as Languir waves) during their travel in the solar corona and interplanetary medium (IPM) (e.g., Ginzburg and Zhelezniakov, 1958; Zheleznyakov, 1970; Mercier, 1975; Melrose, 1980; Pick and Ji, 1986; Saint-Hilaire et al., 2012; Sasikumar Raja and Ramesh, 2013; Reid and Ratcliffe, 2014; Mahender et al., 2020). Type III radio bursts drift from $\sim$200 MHz to 30 MHz in less than 10 seconds and reach 30 kHz in about 1 hour (Suzuki and Dulk, 1985; Bastian et al., 1998; Pick and Vilmer, 2008). The impulsive flares in X-ray and/or $H_\alpha$ wavelengths, exhibit type III radio bursts at their ascending phases (Cane and Reames, 1988). The most detailed and most recent analysis of type III burst properties with their interpretations is given in the article by Reid and Ratcliffe (2014); Mahender et al. (2020) and more generally on solar radio emission (e.g., Dulk, 1985; McLean and Labrum, 1985; Bastian, 1990; Pick and Vilmer, 2008; Gary et al., 2018). On the other hand, type IV bursts are often accompanied by long-duration events observed at EUV or soft X-ray wavelengths and CMEs. Type IV radio bursts are broadband continuum emissions at decimetric and metric wavelengths often associated with CMEs (Pick and Ji, 1986). These bursts can have either stationary or moving sources and various emission mechanisms (Pick and Ji, 1986; Bastian et al., 1998). The main difference between moving type IV (IVm) and stationary type IV (IVs) radio bursts, is that IVm shows an obvious frequency drift, while IVs does not, implying that the radio source of IVm moves outward from the Sun, while IVs is stationary (Pick and Ji, 1986). Type IVs bursts since their discovery, originate from plasma emission (Weiss, 1963; Benz and Tarnstrom, 1976; Salas-Matamoros and Klein, 2020). They prevail upon the presence of non-thermal electrons in the solar corona for several hours in relationship with flare and the liftoff of a CME. Additionally, other phenomena associated with flares and/or CMEs such as small-scale ejecta (e.g., flare sprays, see Vršnak, 2001) or Moreton waves (e.g., Warmuth et al., 2001) can be sources of shocks and may accelerate particles.

Type II, type III and type IV radio bursts play a major role for the benefit of space weather studies because: (i) their observations are used to study the underlying physics of solar eruptive events, (ii) they are often associated with space weather hazards (i.e., flares, CMEs and solar energetic particles (SEPs)) and (iii) they potentially allow the study and tracking of geo-effective disturbances from the Sun through the IP medium to the Earth (e.g, Warmuth and Mann, 2004; Cane and Erickson, 2005; Miteva et al., 2017; Carley et al., 2020). CMEs and solar flares can produce streams of highly energetic particle events which can impact the earth's magnetosphere and ionosphere and affect the performance of radio-communication systems and can even influence the tropospheric weather. Adverse space weather can specifically affect satellite communication, in particular the Global Navigation Satellite Systems (GNSS) or directly damage satellites, while various radiations can be hazardous for astronauts and crew of the flights (e.g,. Carley et al., 2020; Vourlidas et al., 2020). It has been reported in the articles (Cerruti

et al., 2008; Carrano et al., 2009; Muhammad et al., 2015) that the occurrence of an intense SRBs can influence the GPS signal reducing the carrier-to-noise power spectral density ratio (C/N) of all the receivers located in the sunlit hemisphere of the Earth.

It is important to mention here that X-rays, EUV flux and SEPs from the solar flares reach the ionosphere within hours and impact the HF communication while CMEs take 1 to 5 days to arrive at Earth depending on their speeds and direction. This article tends to explain how radio bursts observed from ground play a vital role to predict space weather hazards. The diagnosis of the solar atmosphere by a better understanding of the solar radio emissions is sophisticated by ever-increasing instruments technology. With this advancement, several ground and space telescopes have been built to observe SRBs at a global scale. Space based observations detect interplanetary bursts at $\leq 14$ MHz using WAVES–WIND (Bougeret et al., 1995), and WAVES–STEREO (Kaiser, 2005; Rucker et al., 2005) instruments. Globally distributed ground-based solar radio spectrographs include: Radio Solar Telescope Network (RSTN) operated by US Air-force (Guidice et al., 1981), Hiraiso Radio Spectrograph (HiRAS) in Japan (Kondo et al., 1994), ARTEMIS-IV in Greece (Caroubalos et al., 2001), IZMIRAN in Russia (Gorgutsa et al., 2001), Gauribidanur Low frequency Solar Spectrograph (GLOSS) in India (Kishore et al., 2014), in the decameter wavelength: Nançay Decameter Array (Boischot et al., 1980), Ukrainian T-shape Radio telescope model 2, UTR-2 (Braude et al., 1978; Braude, 1992), Ukrainian Radio interferometer, URAN (Braude et al., 1992; Megn et al., 1997) and many others[1]. The development of the Square Kilometre Array (SKA, Dewdney et al., 2009; Nindos et al., 2019) will open a new opportunity to understand radio-wave propagation. Despite this technological advancement, gaps in data were highly recognized in developing countries, especially African continent (Guhathakurta et al., 2013). In order to tackle these data gaps, the International Space Weather Initiative (ISWI)[2] has contributed to the observation of space weather phenomena through the deployment of ground-based instruments (Haubold et al., 2010). To this end a Compound Astronomical Low-cost Low-frequency Instrument for Spectroscopy and Transportable Observatory (CALLISTO) radio spectrometer (Benz et al., 2005, 2009) has been deployed in different parts of the globe which led to extended CALLISTO (e-CALLISTO). The latter is designed to operate within $10 - 870$ MHz frequency range and probe the solar corona in heliocentric distance raging from 1 solar radius ($R_\odot$) to 3 $R_\odot$ (Pohjolainen et al., 2007). Through collaboration, Rwanda has been the first country in East Africa that acquired a CALLISTO spectrometer.

The current analysis presents the trends of SRBs detected by CALLISTO station in Rwanda and discusses their correlation with solar explosive phenomena which occurred during its first year of operation. The data used covers the period from October 2014 to September 2015 within the Solar Cycle 24 (SC 24). This article is structured as follows. Section 2 describes the observation of the instrument and subsequent compararison with other spectrometers. The methods used to get analyzed data are detailed in section 3. The results and discussions are presented in section 4, while concluding remarks and summary are in section 5.

---

[1]https://www.astro.gla.ac.uk/users/eduard/cesra/?page_id=187

[2]http://www.iswi-secretariat.org

## 2 Observations

Radio burst observations using CALLISTO spectrometer at the University of Rwanda, located in Kigali (1.9441°S, 30.0619°E) began in the year 2014, after the installation of a broadband logarithmic periodic dipole array (LPDA) and low noise amplifier (LNA, http://www.e-callisto.org/StatusReports/status20140718_V0.pdf) as the primary receiving components. CALLISTO Rwanda operates within the frequency range of 45 – 80.9 MHz and is a part of e-CALLISTO network (Benz et al., 2005, 2009). The instrument with a radiometric bandwidth of about 300 KHz, integration time of 1 ms and noise figure of < 3 dB allows it to detect bursts in the order of 30 SFU (solar flux units, $1 \, \text{SFU} = 10^{-22} \text{Wm}^{-2} \text{Hz}^{-1}$). Such a spectrometer is needed to continuously identify the nature of coherent radio emissions and to complement interferometers and array spectrometers (Bastian, 2004). The instrument is well calibrated in time - frequency domain such that it acquires the ability to observe radio spectra as can be seen by any other spectrometer. For instance, Figure 1 illustrates a dynamic spectrum of type III radio bursts observed by the instrument on 24 August 2015 (top) and many others. Among the other spectrometers, the same dynamic spectrum of type III radio burst was observed by Humain Solar radio Spectrometer at Royal observatory of Belgium (ROB) that operates in frequency range 275 MHz to 1495 MHz and is shown as well (bottom). The same event was observed by the radio telescope URAN-2 of Poltava Gravimetrical Observatory of Geophysics Institute of National Academy of Sciences of Ukraine (Poltava) consisting of 512 broadband dipoles. The URAN-2 operates in 8 – 33 MHz frequency range[3]. Furthermore, this burst was also registered by Nançay Decameter Array (NDA) that operates within 10 – 80 MHz frequency range.

## 3 Methods

Radio data used in this study were detected from October 2014 to September 2015 during SC 24. Using quick plots provided by e-CALLISTO network[4] and by manual inspection, we made a cross checking to identify all radio bursts observed by the CALLISTO RWANDA. As a result, a total of 201 radio bursts comprising 4 type II radio bursts, 175 type III radio bursts, and 22 type IV radio bursts were identified within 45 - 80.9 MHz frequency range. The time of observation varies with the seasons of the Sun. Figure 2 shows the spectral overview and the time coverage of the instrument through the year at the station.

Radio bursts are used as diagnostics of the level of solar activity (e.g., Lobzin et al., 2011; Morosan et al., 2015; Salmane et al., 2018), therefore we carefully examined their association with the solar transients using the database from Heliophysics Event Catalogue[5]. The latter gives white light coronagraph data from the Large Angle and Spectrometric Coronagraph (LASCO C2) onboard the Solar and Heliospheric Observatory (SOHO, Brueckner et al., 1995) and the Sun-Earth Connection Coronal and Heliospheric Investigation (SECCHI, Howard et al., 2008) onboard the Solar Terrestrial Relations Observatory (STEREO) Ahead (A) and Behind (B) spacecrafts as well as solar flares. LASCO coronagraphs observe the Sun from $1.1 – 3 \, R_{\odot}$, $2.5 – 6 \, R_{\odot}$ and $4 – 32 \, R_{\odot}$ for coronagraphs 1, 2 and 3, whereas STEREO/SECCHI has two white-light coronagraphs, cor 1 and cor 2 observing the Sun from $1.3 – 4 \, R_{\odot}$ and $2 – 15 \, R_{\odot}$, respectively.

---

[3]http://www.astro.gla.ac.uk/users/eduard/uran/

[4]http://soleil.i4ds.ch/solarradio/callistoQuicklooks/

[5]http://hec.helio-vo.eu/hec/hec_gui.php

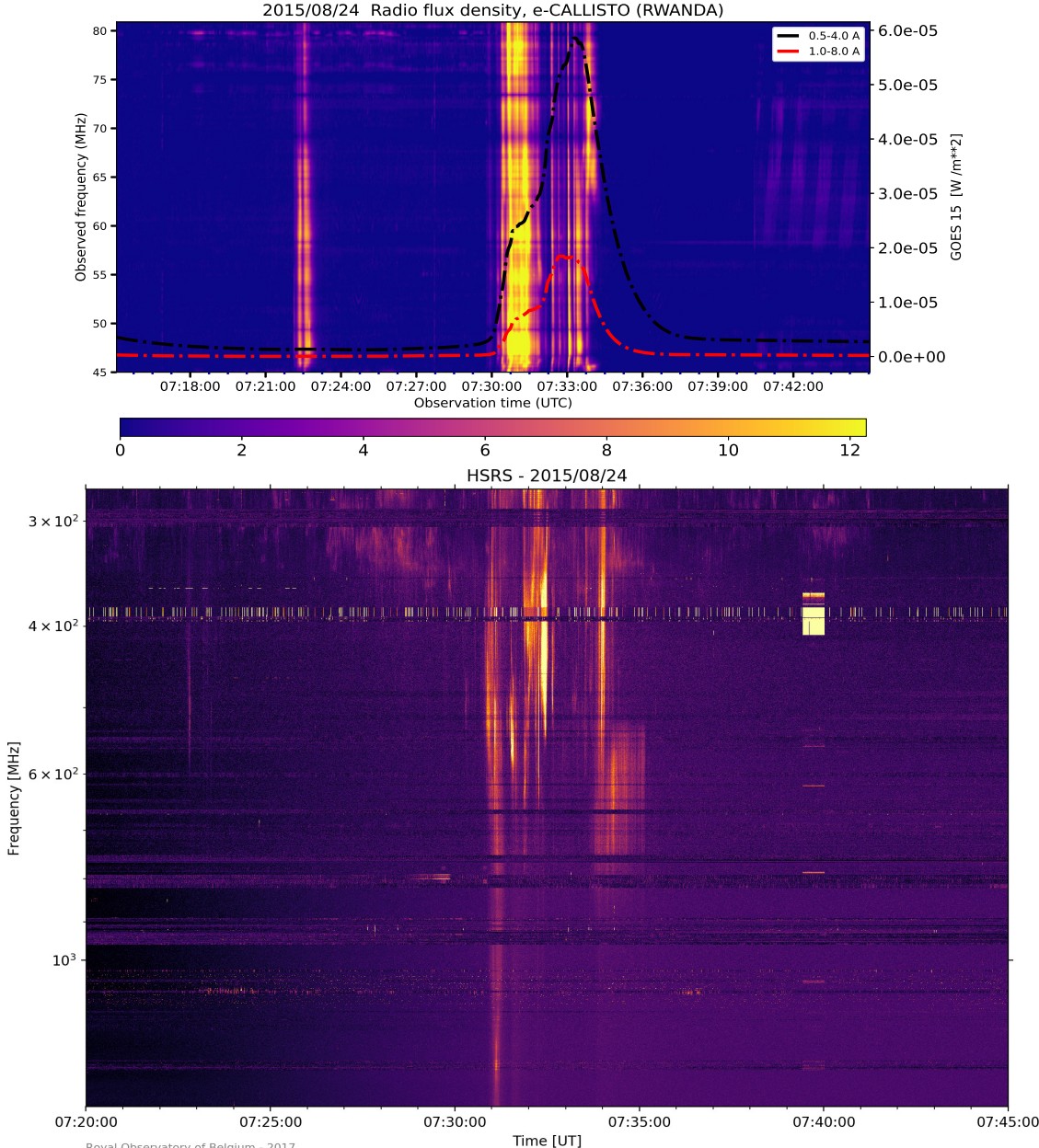

**Figure 1.** Dynamic spectrum of 24 August 2015 type III radio bursts. Top: An isolated type III burst at 07:22 UT followed by a group of type III bursts observed at 07:30 UT by CALLISTO Rwanda. Bottom: a group of type III bursts observed at 07:30 UT by ROB observatory. The one at 07:22 UT occurred during the decaying phase of a C6.7 flare (started, peaked, ended at 07:08 UT, 07:12 UT and 07:15 UT, respectively). A group of type III bursts at 07:30 UT is triggered by a M5.6 flare in its rise phase (started at 07:26 UT, peaked at 07:33 UT and ended at 07:35 UT, respectively. The black and red curves show the GOES X-ray flux.

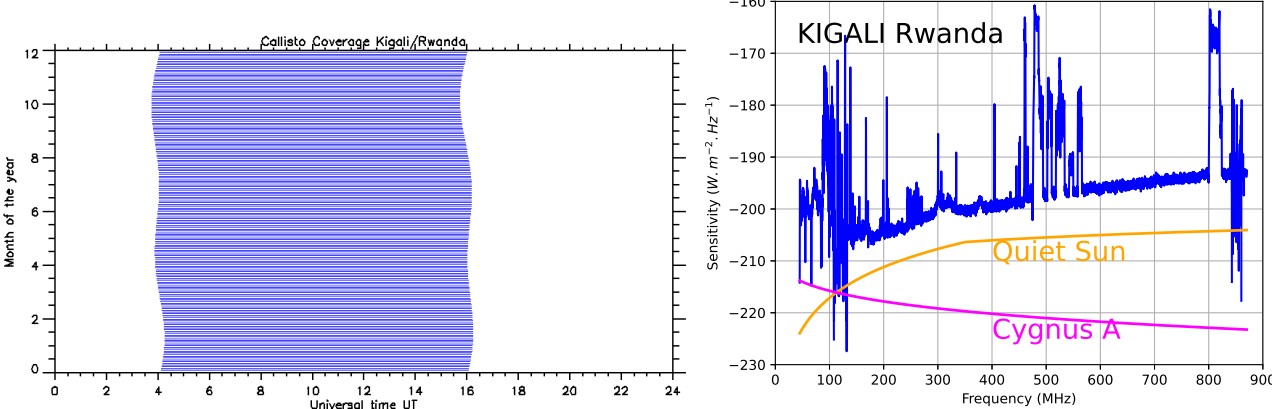

**Figure 2.** The left panel shows the observation coverage by CALLISTO station at University of Rwanda in Kigali. The right panel displays the spectral overview Kigali/Rwanda. Strong peaks near 100 MHz, 500 MHz and 800 MHz are due to FM-radio, digital video broadcast and mobile phone transmitters. Negative peaks with respect to blue baseline are due to saturation of the spectrometer from strong local transmitters. Best frequencies for solar burst observations are 45 MHz-80 MHz, 150 MHz- 450 MHz and 600 MHz-800 MHz due to low interference.

The association of SRBs and CMEs was confirmed upon the time coincidence (a time window of $\sim$ 1 hour of CME from the onset of radio burst was considered). For the events where multiple CMEs occurred at the same time, we checked the source region of the moving bursts from CALLISTO images to associate them with the correct CME. In this study, a burst is considered as flare associated if it appears between its onset and end times. For some type III and type IV bursts, it was not easy to find the driving agent, hence the Atmospheric Imaging Assembly (AIA, Lemen et al., 2012) images on board the Solar Dynamics Observatory (SDO) were used to indicate the possible launch sites. Furthermore, SDO/AIA images enable us to observe jets and their source region with a temporal and spatial resolution (Innes et al., 2011).

## 4  Results and discussions

### 4.1  Statistical analysis of observed radio bursts

The observed bursts and the associated flare properties are split into three tables according to their typical morphology. Table 1 presents 4 intense and well-separated type II radio bursts. Similarly, Table 2 presents type IV bursts identified while a detailed table of type III radio bursts can be accessed from http://www.e-callisto.org/GeneralDocuments/Type_III_radio_bursts_2014_ 2015_kigali.pdf. For each table, the analyzed bursts parameters such as the burst date (yy/mm/dd format), the corresponding onset time, and frequency range are listed in the first four columns, respectively. The next five columns list the associated flare properties: the flare onset time, end times, GOES X-ray flare classes, heliographic coordinates, and active regions (ARs) and

the last column shows associated CME onset, respectively. The analysis shows that all type II radio bursts are associated with

**Table 1.** Type II SRBs observed by e-CALLISTO network, Kigali station.

| | Type II bursts | | | Associated Flares | | | | | CME |
|---|---|---|---|---|---|---|---|---|---|
| Date | Start | Frequency (MHz) | | Time (UT) | | Class | ARs | Locations | Onset |
| | (UT) | Start | Stop | Onset | End | | | | (UT) |
| 2014/11/02 | 09:50:00 | 62.0 | 45 | 09:20 | 10:36 | C4.5 | 2192 | S12W92 | 10:00 |
| 2014/11/05 | 09:49:00 | 80.9 | 45 | 09:26 | 09:55 | M7.9 | 2205 | N15E53 | 10:12 |
| 2015/08/22 | 06:52:23 | 75.0 | 45 | 06:39 | 06:59 | M1.2 | 2403 | S14E09 | 07:12 |
| 2015/08/28 | 06:36:00 | 67.0 | 47 | 06:17 | 06:38 | C4.5 | 2403 | S15W71 | 06:36 |

solar flares, ∼37% of the type III and ∼13% of type IV radio bursts detected are flare associated. Findings are compared with the results obtained in the previous similar studies using other CALLISTO stations such as Mahender et al. (2020) who found that 426/1531 ($\sim 28\%$) type III bursts were flare associated and Ndacyayisenga et al. (2021) who inferred that 65% of 698 type III radio studied were flare associated. The remaining non-flare associated type III radio bursts may be due to small-scale reconnection events present in the solar corona. Furthermore, the association of these bursts with CMEs was traced out. It is

observed that all analyzed type II radio bursts are associated with CMEs. The obtained results agree with the recent study by Umuhire et al. (2021) who found that among the 365 type II bursts reported on SWPC, only 84 (23%) were not associated with CMEs. The study found that ∼44% of type III radio bursts are accompanied by CMEs. The majority ($\sim 82\%$) of type IV radio bursts are associated with CMEs which is consistent with the results obtained by Kumari et al. (2021) who analyzed a set of 446 type IV radio bursts and found that 81% of them are temporally and spatially correlated with CMEs.

Among the type II bursts detected by the instrument, a case study of the 5 November 2014 event is taken. Figure 3 (a) shows a type II radio spectrum recorded on 5 November 2014 by CALLISTO spectrometer based in Rwanda. Type II radio burst presented in the figure shows both fundamental and harmonic band split structures. The fundamental band started at 09:48:43 UT with 80.0 MHz and ended at 09:53:32 UT with 47.0 MHz while the harmonic band started at 09:51:00 UT with 80.9 MHz and ended at 09:56:12 UT with 59.0 MHz which corresponds to the drift rates of 0.11 MHz/s and 0.07 MHz/s, respectively.

The 5 November 2014 type II burst was also detected by a solar radio spectrograph located at the Learmonth solar observatory, Australia among many other instruments as shown in Figure 3(b). The spectrum obtained by the two different instrument are of good quality for spectral analysis. This event was associated with a M7.9 X-ray flare (refer to Figure 3(c)) that started at 09:26 Ut and ended at 09:55 UT in the NOAA 12205 active region located at N15E53 on the solar disk. The 5 November 2014 burst was also associated with a partial halo CME with a linear speed of 386 km/s which was only detected by LASCO (C2

and C3). An illustrating CME image is shown in Figure 3 (d).

      A small fraction of type III bursts is found to be associated with impulsive flares. Observed radio bursts with no direct connection to impulsive flares may have originated at $H_\alpha$ ejecta, X-ray footpoints and X-ray or/and extreme-UV (EUV) jets (Alissandrakis et al., 2015, and references therein). The 3 November 2014 is selected among the type III bursts reported in the table maintained at http://www.e-callisto.org/GeneralDocuments/Type_III_radio_bursts_2014_2015_kigali.pdf because a

maximum number of type III radio bursts were detected on this day. Out of 12 well-separated type III radio bursts reported, only one is triggered by solar flare. With the help of images provided by AIA/SDO, a region of open magnetic field lines as the signature of the electron stream responsible for type III radio bursts is indicated. This observation is consistent with earlier studies inferred that type III radio bursts are commonly associated with jets in extreme Ultraviolet (EUVI) and x-rays (e.g., Bain and Fletcher, 2009; Klassen et al., 2011; Krucker et al., 2011; Klassen et al., 2012) with typical electron beams coinciding the path as the jets. Figure 4 shows a sample of the SDO/AIA images using 171 Å bandpass of the Sun on 3 November 2014. From Figure 4, it is seen that the position of the open magnetic field lines is stationary from 06:55:15 UT to 10:04:05 UT, and hence one can believe that there is a repetitive type III radio emission originating from the same source (e.g., Chifor et al., 2008). A similar trend was observed on the 9 type III radio bursts observed in the absence of solar flares on 9 July 2015. On the other hand, Figure 5 (a) displays a dynamic spectrum of type II radio burst followed by type III radio burst of 2 November 2014, detected by CALLISTO station in Rwanda. These two types of radio bursts were associated by the C4.5 flare that started and ended at 09:20 to 10:36 UT, respectively. Figure 5 (b) shows the same type II burst detected by San-Vito (SVTO) radio spectrograph on board the RSTN located in Italy. RSTN/SVTO couldn't detect the followed type III burst. Even though it has a low quality (Umuhire et al., 2021), this reflects the capability of CALLISTO to detect solar radio burst in its designed frequency. The CME associated with these events was off the Sun-Earth line. Although the small fraction of type III radio bursts, we have plotted the heliographic longitudes and latitudes of the associated solar flares as indicated in Figure 6. It is seen that the distribution of flares associated with type III radio bursts originate near the equator ($\pm 30°$). This result is consistent with the findings by Mahender et al. (2020) who found that the analyzed 426 type III bursts associated with solar flares originating close to the equator (i.e. heliographic latitudes $\pm 23°$).

The analysis continues with Subsequent type IV radio bursts identified in the study period. Table 2 lists all type IV radio bursts observed by the instrument and their association with the solar phenomena. From this table, it can be seen that type IV bursts are highly associated with CMEs which indicates that, their presence can be used to map the location of trapped electrons and studying the CME kinematics during the phases of eruption processes (Kumari et al., 2021).

Table 2: Type IV SRBs observed by CALLISTO spectrometer, Kigali station.

| | Type IV bursts | | | | Soft X-ray flares | | | | | CME |
|---|---|---|---|---|---|---|---|---|---|---|
| $N^0$ | Date | Start | Frequency (MHz) | | Start | End | Location | Class | ARs | Onset |
| | yyyy/mm/dd | (UT) | Start | Stop | (UT) | (UT) | position | | | (UT) |
| 1 | 2014/10/30 | 10:11:45 | 66.0 | 45 | 09:54 | 10:06 | S12W92 | M1.2 | 2192 | .... |
| 2 | 2014/11/06 | 08:27:00 | 78.0 | 48 | .... | .... | .... | .... | .... | 08:00 |
| 3 | 2014/11/14 | 08:23:00 | 73.0 | 45 | .... | .... | .... | .... | .... | 08:12 |
| 4 | 2014/11/25 | 15:56:35 | 80.9 | 45 | .... | .... | .... | .... | .... | .... |
| 5 | 2014/12/05 | 16:06:00 | 80.9 | 45 | .... | .... | .... | .... | .... | 15:24 |
| 6 | 2015/01/01 | 11:00:00 | 78.0 | 45 | .... | .... | .... | .... | .... | 10:36 |
| 7 | 2015/01/09 | 09:34:10 | 80.0 | 48 | .... | .... | .... | .... | .... | 08:24 |

| | | | | | | | | | | |
|---|---|---|---|---|---|---|---|---|---|---|
| 8 | 2015/01/12 | 09:07:32 | 78.0 | 45 | .... | .... | .... | .... | .... | .... |
| 9 | 2015/01/12 | 10:31:00 | 75.0 | 45 | .... | .... | .... | .... | .... | .... |
| 10 | 2015/01/13 | 09:14:00 | 80.9 | 45 | .... | .... | .... | .... | .... | 09:12 |
| 11 | 2015/01/17 | 15:20:22 | 80.9 | 45 | .... | .... | .... | .... | .... | 15:12 |
| 12 | 2015/01/21 | 11:26:10 | 80.9 | 45 | .... | .... | .... | .... | .... | 11:24 |
| 13 | 2015/01/27 | 08:31:30 | 80.9 | 45 | 07:13 | 08:27 | S16W16 | C2.1 | 2275 | 08:00 |
| 14 | 2015/01/30 | 14:26:30 | 80.9 | 45 | .... | .... | .... | .... | .... | 13:48 |
| 15 | 2015/02/06 | 08:44:00 | 80.9 | 45 | .... | .... | .... | .... | .... | 08:24 |
| 16 | 2015/02/10 | 09:27:30 | 72.0 | 45 | .... | .... | .... | .... | .... | 09:24 |
| 17 | 2015/02/24 | 10:13:00 | 78.0 | 45 | 09:54 | 11:53 | S21E87 | C1.4 | .... | 10:00 |
| 18 | 2015/04/16 | 06:43:10 | 80.9 | 45 | .... | .... | .... | .... | .... | 06:36 |
| 19 | 2015/04/24 | 10:52:25 | 80.9 | 45 | .... | .... | .... | .... | .... | 10:48 |
| 20 | 2015/04/29 | 06:11:56 | 77.0 | 45 | .... | .... | .... | .... | .... | 05:36 |
| 21 | 2015/05/28 | 12:14:57 | 73.0 | 45 | .... | .... | .... | .... | .... | 12:00 |
| 22 | 2015/07/08 | 18:22:50 | 68.0 | 45 | .... | .... | .... | .... | .... | 12:12 |

We considered a case of January 2015 where a total of 9 type IV radio bursts were detected and we analyse 6 of them. Figure 7 presents the magnetic loops embedded in the solar corona routinely the site of trapped electrons responsible for Type IV emissions. The figure displays a sample of six AIA/SDO images in a 171 Å band filter corresponding to the dates of type IV radio bursts in January 2015. Recent results show that the type IV radio bursts coincide with the decaying phase of flares and/or triggered by post-eruption loops (e.g., Morosan et al., 2019; Kumari et al., 2021). Moreover, these bursts may lack both association with flares and CME eruptions (Table 2).

## 4.2 Space weather monitoring

In this article, we attempt to explain how SRBs monitoring and analysis can be used to diagnose space weather hazards. SRBs (type II, III and type IV) can be considered as early signatures to space weather phenomena since they are often observed in association with flares and/or CMEs which are major drivers of space weather hazards including geomagnetic storms (e.g., Warmuth and Mann, 2004). Thereby, the detection of SRBs using ground based instrumentation can serve as the advance warning of incoming associated solar transient events and provide insights towards predicting subsequent space weather hazards. For example, HF communications and the ionosphere can be disrupted by the X- ray and EUV wavelengths along with the solar energetic particles that may reach Earth within a few hours after onset of solar flares and CMEs (e.g., Marqué et al., 2018). Type IV bursts are attributed to electrons trapped in closed field lines in the post-flare arcades and have high degree of association with solar energetic particle events (White, 2007). Thus they can be used as extreme solar radiation precursors. An example is an extreme UV radiation on 13 January 2015 from the flare which ionized Earth's atmosphere and caused radio

blackouts in different parts of the Earth. This hazard happened while our instrument detected two type IV bursts on 12 January 2015 and one on the next day. Affected areas are shown on top of Figure 8. Another example of radio blackout was observed on 12 March 2015 when three stations of e-CALLISTO network observed long-lived type IV radio burst by KRIM station (Radio Astronomy Laboratory of CrAO, Ukraine) from 04:10 UT to 04:40 UT followed by type III radio burst at 04:43 UT (KRIM) and at 11:25 UT (Rwanda), respectively. The disturbances caused by X-ray enhancement are illustrated in the bottom of Figure 8. There are more than 150 e-CALLISTO spectrometers installed around the world and operate simultaneously. This makes the e-CALLISTO network the second to the USAF RSTN to cover the 24 hour observation (e.g., Benz et al., 2009; Zucca et al., 2012; Sasikumar Raja et al., 2018). The observations from a single station can be limited by the sensitivity of the antenna. Despite technical issues, a burst can be viewed by more than one spectrometer and this determines the uniqueness of the e-CALLISTO network. With the help of dynamic spectra provided by this network, it has been shown in a recent related analysis (Ndacyayisenga et al., 2021) that space weather can continuously be monitored at a large scale. Therefore a combination of information provided by each station as part of the e-CALLISTO network may contribute to revealing hidden features in space weather diagnostics.

## 5   Summary and conclusion

In this study, we analyzed 201 SRBs detected by the CALLISTO spectrometer installed in Kigali, Rwanda. The obtained results show that during its first year of operation, 4 type II, 175 type III and 22 type IV radio bursts were detected in the frequency range 45–80.9 MHz. Analyzed data indicate that all detected type II radio bursts are flare associated; $\sim 37\%$ of observed type III bursts are triggered by impulsive flares. We found that the remaining non-flare related type III bursts might have been triggered by small-scale features or weak energy events present in the solar corona according to the literature and with the help of AIA/SDO images in the 171 Å bandpass. All detected type II radio bursts are associated with CMEs while only $\sim 44\%$ of type III are found to be accompanied by CMEs. Furthermore, we observed that the majority ($\sim 82\%$) of detected type IV radio bursts are accompanied by CMEs. Major findings from this study clearly indicate the importance of analysing SRBs observed from ground instrumentation in the study of space weather. More instruments for a wide coverage and regular maintenance of such instruments is recommended for a better monitoring and prediction of space weather to supplement existing monitoring using space borne instruments as well as radio interferometer arrays.

*Author contributions.* T. Ndacyayisenga, Jean Uwamahoro and A. C Umuhire conceived of the presented idea and the design of the study. T. Ndacyayisenga manually gathered the data used. C. Monstein helped in programming for data analysis. Analysis and intrepretation of the results are done by T Ndacyayisenga and A. C Umuhire who later drafted the manuscript. This manuscript is critically reviewed by Jean Uwamahoro and C. Monstein for intellectual content. Oversight, leadership responsibility for the research activities, mentorship are the inputs from Jean Uwamahoro.

*Competing interests.*    The authors declare that the research was conducted in the absence of any commercial or financial relationships that could be created as a potential conflict of interest.

*Acknowledgements.*    This work was partially supported by the International Science Program (ISP) through the Rwanda Astrophysics, Space and Climate Science Research Group (RASCSRG). We thank FHNW, Institute for Data Science in Brugg/Windisch, Switzerland for hosting the e-Callisto network. SOHO/LASCO; NOAA; GOES; SWPC and Royal Observatory of Belgium (Brussels); RSTN, AIA/SDO; STEREO
A/B; the Helioviewer (http://helioviewer.org); Poltava Gravimetrical Observatory of Geophysics (Ukraine) and Nançay radio observatory (France) to make their data available online. This study also used version 2.1.5 of the SunPy open source software package (SunPy Community et al., 2020). (the team leaders E. Yizengaw & K. Groves) for valuable discussion about part of the result that are included in this paper.

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

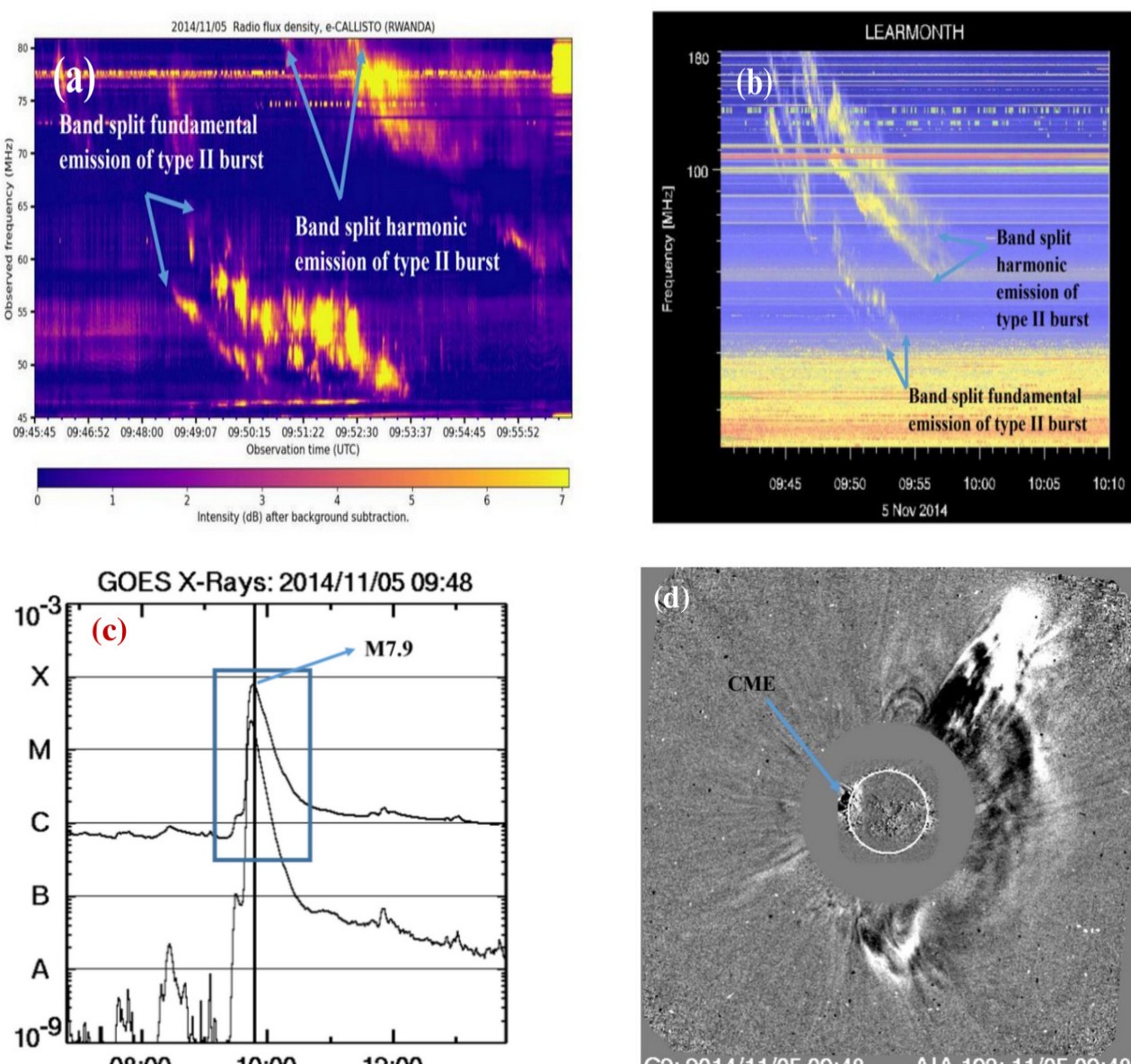

**Figure 3.** (a) The dynamic spectrum of the 5 November 2014 type II radio burst detected by CALLISTO station in RWANDA. (b) The 5 November 2014 type II burst observed by the Learmonth radio observatory. The event has fundamental- harmonic (F-H) structure, each with a band split due to the magnetic field. The fundamental band started at 09:48:43 UT with 80.0 MHz and ended at 09:53:32 UT with 47.0 MHz while the harmonic band started at 09:51:00 UT with 80.9 MHz and ended at 09:56:12 UT with 59.0 MHz which corresponds to the drift rates of 0.11 MHz/s and 0.07 MHz/s, respectively. (c) A M7.9 x-ray flare associated with the 5 November 2014 type II burst as detected by GOES satellite. Its starting and ending times are 09:26 UT and 09:55 UT, respectively. (d) LASCO C2 difference image shown by blue arrow. The starting time of type II burst, the flare onset and the CME are close as shown in the images.

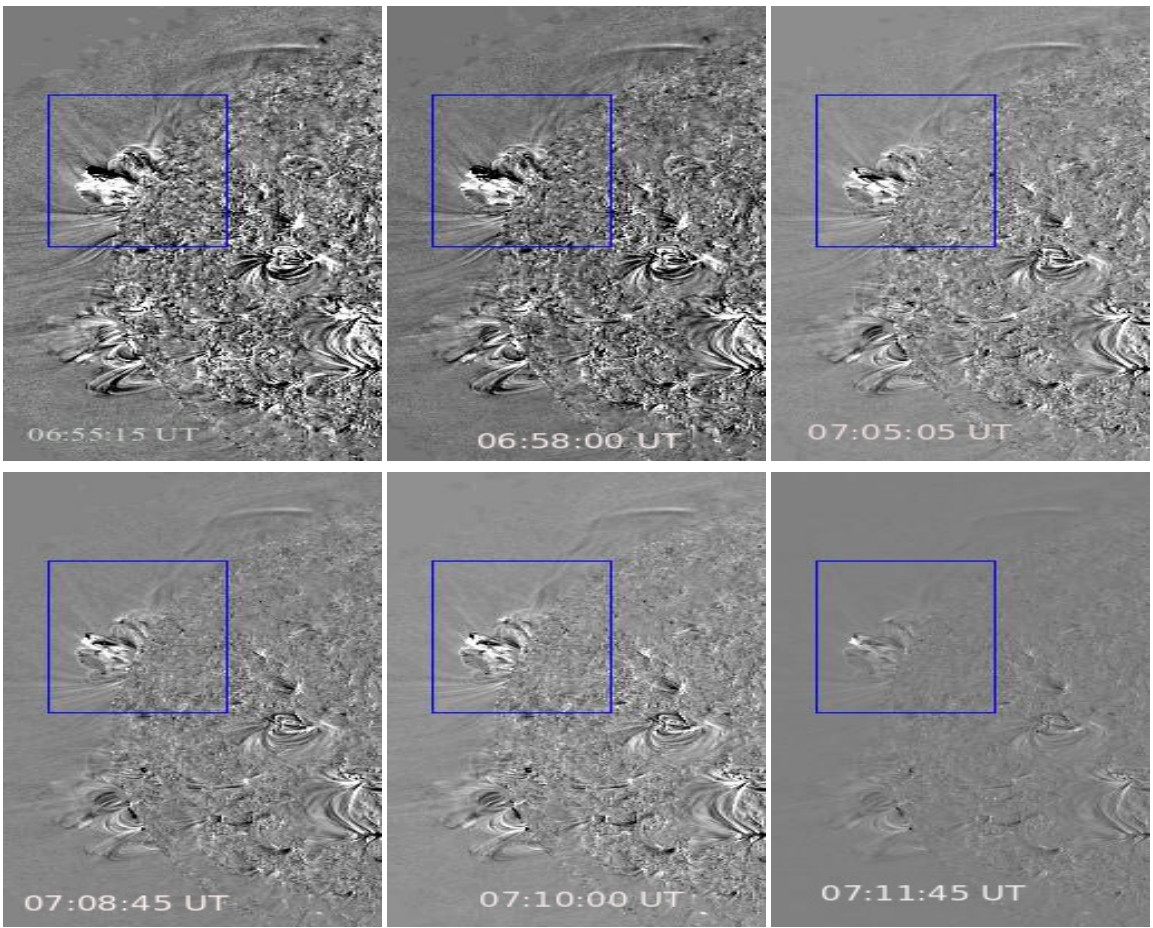

**Figure 4.** AIA/SDO portion images of the Sun using 171Å bandpass on 3 November 2014. These are running difference images corresponding to 6 type III radio bursts among 11 bursts observed on 3 November 2014 from 06:55:15 UT to 10:04:05 UT, respectively. This figure shows the region of open magnetic field lines enclosed by blue rectangles which are the path of electron beams emitting type III radio bursts.

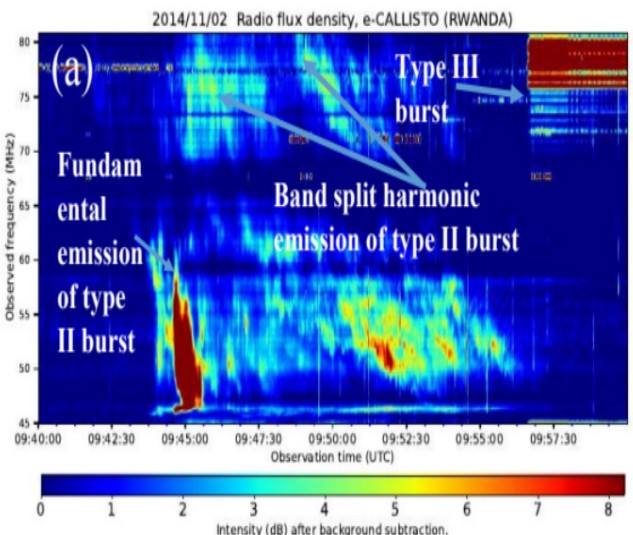
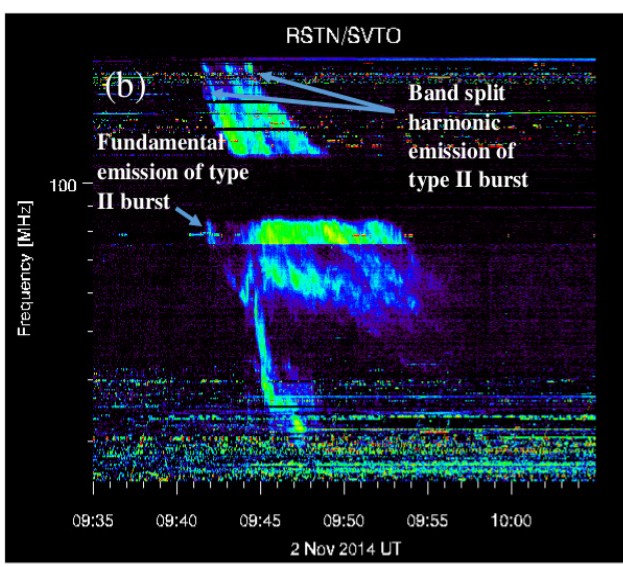

**Figure 5.** (a) The dynamic spectrum of the 2 November 2014 type II burst followed by type III burst detected by CALLISTO station in RWANDA. (b) The same dynamic spectrum the 2 November 2014 burst detected from the Radio Solar Telescope Network (RSTN), San-Vito (SVTO) station in Italy. The observed type III is of low quality and it was only detected by CALLISTO which proves its capacity to detect radio emissions.

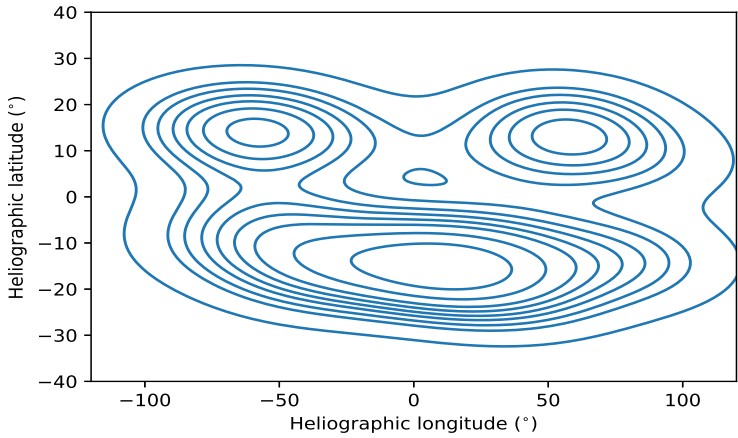

**Figure 6.** Heliographic longitude vs. heliographic latitude of flares that have triggered the type III radio bursts.

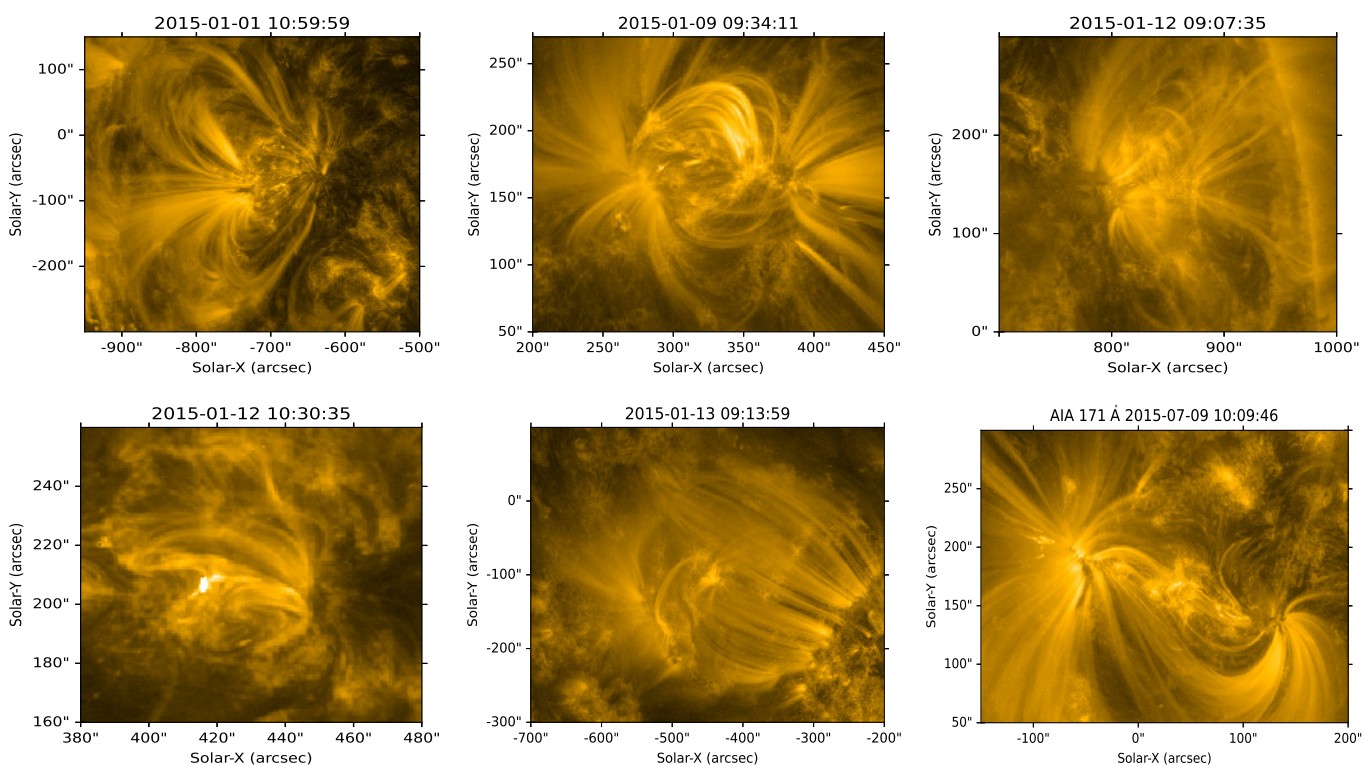

**Figure 7.** The magnetic-loops embedded in the solar corona indicating the potential site of trapped electrons responsible for Type IV emissions. The figure displays a sample of six AIA/SDO images in 171 Å band filter corresponding to the dates of type IV radio bursts in January 2015.

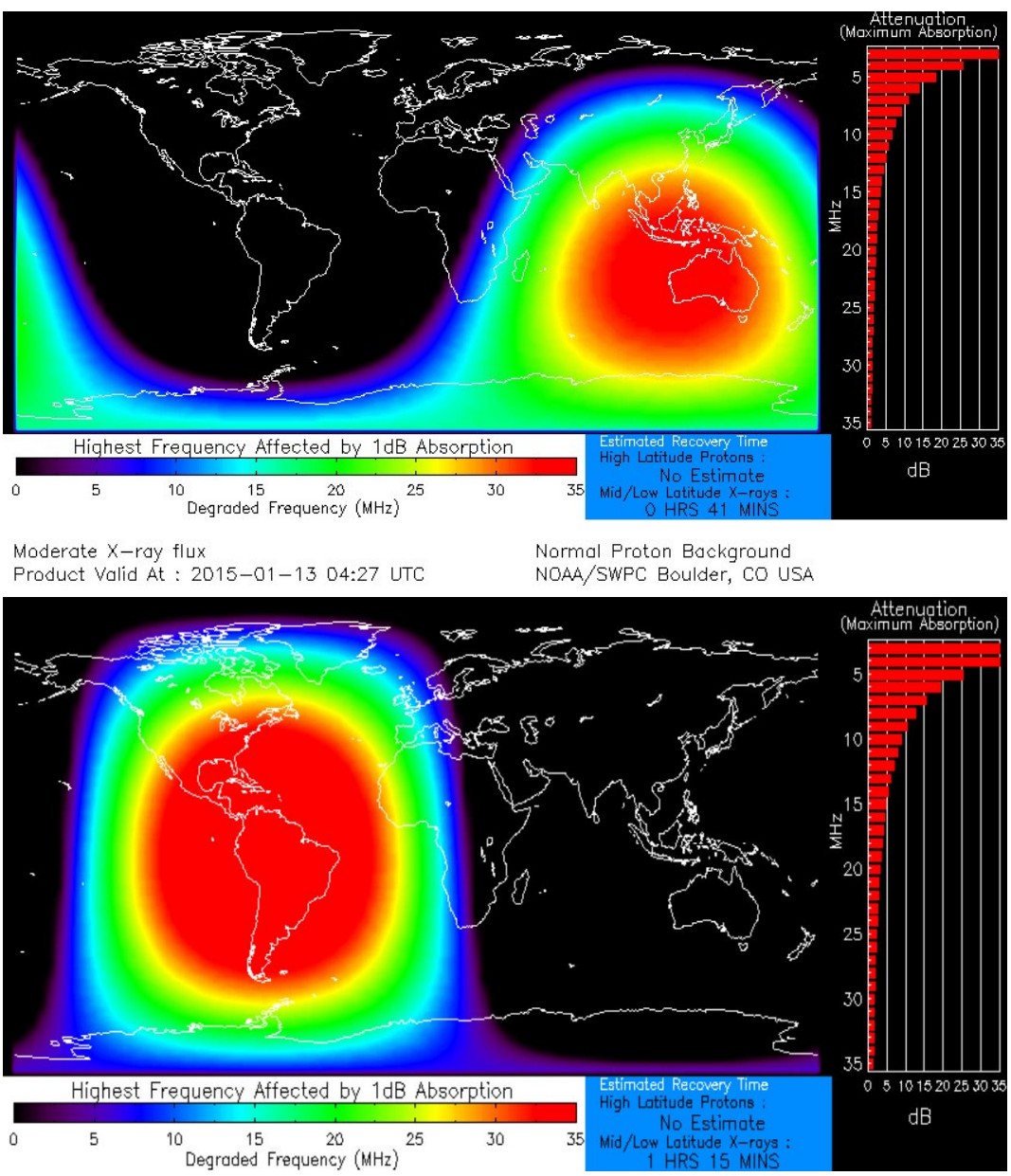

**Figure 8.** Maps with radio blackout on 13 January 2015 (top) and on 12 March 2015 (bottom). Creditted to SWPC.