# Peer review of "Space Weather Study through Analysis of Solar Radio Bursts detected by a Single Station CALLISTO Spectrometer"

_Annales Geophysicae, 2021_

## Referee Comment (RC1)

Review angeo-2021-26

**Space Weather Study through Analysis of Solar Radio Bursts detected by a Single Station CALLISTO Spectrometer**

by T. Ndacyayisenga et al.

The paper is a summary of results of solar radio bursts analysis observed by the e-Compound Astronomical Low cost Low-frequency Instrument for Spectroscopy and Transportable Observatory (e-CALLISTO), comprising the period Oct 2014 – Sept 2015, at the University of Rwanda, College of Education. Main motivation of this study is formulated to use radio burst data to check their correlation with solar activities.

In Chapter 1 – „Introduction" a (too) short overview is presented on the morphologies of the various types of solar radio bursts, indicating the registration of 5 Type II bursts, 175 Type III bursts, and 22 Type IV bursts within the given period of observation. The data base of Type II and Type IV bursts is rather scarce, apparently owed to the limited sensitivity of the used antennas. At this point a more detailed description of the applied instruments (antennas and backend facilities) is strongly recommended.

Among the mentioned spaceborne and ground-based radio observatories (lines 34 – 39) reference should be given to renowned radio telescopes of Nancay (France) and Charkiv and Poltava (Ukraine), resp., which provide with their huge arrays excellent observations in the decameter range.

In Chapter 2 – „Observation and methods" a comparison and supplimentary investigation of the e-CALLISTO results with observations of other stations would enable a better classification and judgement of the quality of the e-CALLISTO results obtained at the College of Education, University of Rwanda. Among other stations the use of the world-wide distributed e-CALLISTO network just offers this opportunity.

In Chapter 3 – „Results and discussions" the obtained radio data are listed in tables (including an online-link for the Type III burst data which however – checked on different days - is unavailable !) with the „associated" solar phenomena (Table 1 and Table 3). In this respect the terms „associated" and „related" (line 72) need a clear definition. Table 2 is trivial and its content can be described in not more than two short sentences.

The solar radio burst observed on Aug 22, 2015, had a frequency stop at 45 MHz (according to Table 1) whereas in the text (line 82) 46 MHz are given.

Figure 2c visualizes a CME image where the red line is considered as the CME shock height. This needs a clear physical justification.

Figure 3 shows running difference images with the last image taken at 10:04:05 UT, correctly cited in the figure legend, whereas in line 103 the time 10:05 UT is mentioned.

As the title of the paper indicates ( „Space Weather Study …") this branch of space physics comprises the varying conditions within the solar system, caused by the sun, including near-Earth conditions, e.g. in the magnetosphere and ionosphere. At least some examples of strong solar radio emission observed by e-CALLISTO and connected with solar plasma phenomena (flares, CMEs, shock waves) and their impact on Earth would clearly demonstrate the „main motivation of this study" as mentioned above.

It should also be pointed out in what phase of solar cycle the observations have been performed in order to evaluate the number, the kind and strength of radio bursts and their connection to solar phenomena.

A few typoes should be corrected:

The paper title should correctly be written: „CALLISTO" (missing „I")

Abstract, line 2: „Spectroscopy" (capital „S")

Line 47: „Education" (capital „E")

Table 2 legend: „Summary" (delete one „m")

Line 96: „remaining"

Line 99: „sample"

Line 119: „It can be seen"

Abbreviations should be explained:

Line 87: „Rs", Line 142: „SRBs"

General comments, Review evaluation:

In terms of innovative contents and originality the paper is of low quality. By consideration of the observational potential of the  e-CALLISTO network  a better and certainly valuable investigation is possible. In view of the above said the paper needs a major, strong and decisive revision and improvement. The referee is willing to re-evaluate the revised version.

---

## Author Comment (AC1)

**Title:** **Space Weather Study through Analysis of Solar Radio Bursts detected by a Single Station CALLISTO Spectrometer.**

**Answers to the referee #1 comments**

Firstly, we thank the referee for providing useful comments on our manuscript. Following the referee's comments, we have carefully gone through the manuscript and revised it. **For the sake of convenience, the newly added text in the manuscript is highlighted in boldface**. Herewith, we provide the answers and/or explanations to the referee's comments:

Comment #1

**The paper is a summary of results of solar radio bursts analysis observed by the e-Compound Astronomical Low cost Low-frequency Instrument for Spectroscopy and Transportable Observatory (e-CALLISTO), comprising the period Oct 2014 – Sept 2015, at the University of Rwanda, College of Education. Main motivation of this study is to use radio burst data to check their correlation with solar activities.**

**Response**

We  thank the referee for his/her suggestion.  The objective is reformulated.

Comment #2

**In Chapter 1 – "Introduction" a (too) short overview is presented on the morphologies of the various types of solar radio bursts, indicating the registration of 5 Type II bursts, 175 Type III bursts, and 22 Type IV bursts within the given period of observation. The data base of Type II and Type IV bursts is rather scarce, apparently owed to the limited sensitivity of the used antennas. At this point a more detailed description of the applied instruments (antennas and backend facilities) is strongly recommended.**

**Response**

The introduction has been revised. The review of various types of solar radio bursts is limited and is beyond the scope of the current article. We have provided references for detailed literature and the introduction is as follows:

'*About eight decades ago, solar radio bursts (SRBs) were classified into five types based on their morphologies and drift rates (Wild, 1950). From the meter to decimeter range, characteristic burst signatures correspond to well-identified physical processes, such as shock waves (type II bursts, Nelson and Melrose (1985); Cairns et al. (2003); Ganse et al. (2012)), electron beams streaming along open magnetic field lines (type III bursts, Lin et al. (1981, 1986)), or electron populations trapped in eruptive flux ropes and post-flare loops (type IV bursts, Nindos et al. (2008)). Type II radio bursts are the bright radio emissions often associated with CMEs and characterized by a slow frequency drift rate ($\leq -1\,MHz\,s^{-1}$) (McLean and Labrum, 1985; Nelson and Melrose, 1985). They are excited by magneto-hydrodynamics (MHD) shocks in the solar atmosphere (Nelson and Melrose, 1985; Cliver et al., 1999; Nindos et al., 2008, 2011; Vršnak and Cliver, 2008). MHD shocks are driven by both flares and CMEs in the solar atmosphere (Nindos et al., 2008). The type II radio emissions are generated at the fundamental and second harmonic of the local plasma frequency. The type III radio bursts are the intense, frequently observed, and fast drifting bursts from high to low frequencies in the dynamic spectra. These bursts usually come from active regions (Saint-Hilaire et al., 2013) and they are generated when solar flares send electron beams streaming into the heliosphere via plasma mechanism (Ginzburg and Zhelezniakov, 1958). Type III radio bursts typically appear as isolated bursts that last in 1–3 s, in groups that last in 10 minutes and as storms that last a few hours. The impulsive flares in X-ray and/or H α frequencies exhibit type III radio bursts at their ascending phases (Cane and Reames, 1988). The most detailed and most recent analysis of type III burst properties with their interpretations is given in the article by Reid and Ratcliffe (2014) and more generally on solar radio emission (e.g, Dulk (1985); McLean and Labrum (1985); Bastian (1990); Pick and Vilmer (2008); Gary et al. (2018)).*

*On the other hand, type IV bursts are often accompanied by long-duration events observed at EUV or soft X-ray wavelengths and coronal mass ejections (CMEs). Type IV radio bursts are broadband continuum emissions at decimetric and metric wavelengths often associated with CMEs (Pick, 1986). These bursts can have either stationary or moving sources and various emission mechanisms (Bastian et al., 1998). Stationary type IV radio bursts (hereinafter referred to as type IVs) since their discovery, originate from plasma emission (Weiss, 1963; Benz and Tarnstrom, 1976; Salas-Matamoros and Klein, 2020). Type IVs prevail over the presence of non-thermal electrons in the solar corona for several hours in relationship with flare and the liftoff of a CME.'*

*The standard situation is that type III radio bursts are more intense and frequent emission from the Sun than type II and type IV bursts. The number of events reported herein has no relationship with brief introduction given. For example in Ndacyayisenga et al., 2021, we reported a total of 12971 type III radio bursts from Space Weather Prediction Centre (SWPC) occurred between 2010 and 2017, while in the same range, only 426 type II radio bursts have occurred.*

***Regarding the instrument****: CALLISTO instrument was designed in the framework of IHY2007 and the idea of providing a cheap instrument to support developing countries in solar radio astronomy. Total cost for such a telescope was decided to be in the order of US2000\$ which every interested institute can afford. This financial constraint was only possible by selecting a cheap antenna (LPDA) with main beam gain of ~6 dB and a low cost frequency agile spectrometer as a back-end. Any high sensitive digital back-end with similar frequency range would cost in the order of 100 times more than a CALLISTO spectrometer. Given this low antenna gain, selected frequency range 45 MHz - 80 MHz, low noise amplifier with ~2 dB noise figure, 300 KHz radiometric spectrometer bandwidth and 1 ms integration leads to a system sensitivity during transit of the Sun in the order of 22 sfu to 66 sfu. Thus, such a low cost system can never see weak bursts, it can only detect strong bursts during Sun transit +/- 3 hours. A CALLISTO based radio telescope can never detect quiet Sun unless the antenna (LPDA) is replaced by a parabolic dish in the order of at least 5m diameter and an appropriate tracking system. Nevertheless a CALLISTO system allows one to step into solar radio astronomy and study dynamic radio spectra.*

 Comment #3

**Among the mentioned spaceborne and ground-based radio observatories (lines 34 – 39) reference should be given to renowned radio telescopes of Nancay (France) and Charkiv and Poltava (Ukraine), resp., which provide with their huge arrays excellent observations in the decameter range.**

   **Response**

We thank the referee for the suggestion. The recommended instruments have been added to the appropriate lists.

Comment #4

**In Chapter 2 – "Observation and methods" a comparison and supplementary investigation of the e-CALLISTO results with observations of other stations would enable a better classification and judgement of the quality of the e-CALLISTO results obtained at the College of Education, University of Rwanda. Among other stations the use of the world-wide distributed e-CALLISTO network just offers this opportunity.**

**Response**
The aim of the current study is to demonstrate the possibility of a single station to the contribution of the space weather studies using observations of solar radio bursts. To test the efficiency of the CALLISTO with any other spectrometers may shift the purpose of this paper.

Using the e-CALLISTO data, it is reported in Ndacyayisenga et al., 2021 that the e-CALLISTO observed 698 intense type III radio bursts among 12971 bursts from Space Weather Prediction Center (SWPC) within the range of 2010 - 2017. In the similar range only 426 type II radio bursts are on SWPC. Similar analysis by Umuhire et al.,2021 found that among 365 type II bursts, only 107 type II bursts could be detected by e-CALLISTO spectrometers.

Comment #4

**In Chapter 3 – "Results and discussions" the obtained radio data are listed in tables (including an online-link for the Type III burst data which however – checked on different days - is unavailable!) with the "associated" solar phenomena (Table 1 and Table 3). In this respect the terms "associated" and "related" (line 72) need a clear definition. Table 2 is trivial and its content can be described in not more than two short sentences.**

**Response**

We thank the referee for pointing out these issues. We have included in the text a correct link which is also given here:
http://www.e-callisto.org/GeneralDocuments/Type_III_radio_bursts_2014_2015_kigali.pdf
 The term 'related' was removed and kept 'associated' which has a clear meaning in the context. However, table 2 was removed.

      Comment #5

**The solar radio burst observed on Aug 22, 2015, had a frequency stop at 45 MHz (according to Table 1) whereas in the text (line 82) 46 MHz are given.**

**Response**

The difference between the two values arose from typing. The correct value is 45 MHz and we have corrected the text accordingly.

      Comment#6

**Figure 2c visualizes a CME image where the red line is considered as the CME shock height. This needs a clear physical justification.**

      **Response**

The CME shock height was taken as the radius of the circle fitted to the outermost part of the disturbance; the assumption is  that the disturbance expands spherically above the solar surface. Hence the CME shock height is shown by the red line. More details on the method used can be found in Gopalswamy et al., 2013.

      Comment # 7

**Figure 3 shows running difference images with the last image taken at 10:04:05 UT, correctly cited in the figure legend, whereas in line 103 the time 10:05 UT is mentioned.**

**Response**

We thank the referee for finding it out. The mistake is corrected.

Comment # 8

**As the title of the paper indicates ( "Space Weather Study ...") this branch of space physics comprises the varying conditions within the solar system, caused by the sun, including near-Earth conditions, e.g. in the magnetosphere and ionosphere. At least some examples of strong solar radio emission observed by e-CALLISTO and connected with solar plasma phenomena (flares, CMEs, shock waves) and their impact on Earth would clearly demonstrate the "main motivation of this study" as mentioned above.**

**Response**

The authors thank the referee for raising this issue. It is clear that this part was missed out which led to the title to be a bit irrelevant.

We have taken care of this section and included a discussion with supporting examples. In this regard: Using space weather website, it is seen that the flare associated with type II bursts given in Table 1 have given rise to an alert of incoming aurora phenomena in polar regions. Type IV bursts are attributed to electrons trapped in closed field lines in the post-flare arcades and have high degree association with solar energetic particle events (White, 2007). Thus they can be used as extreme solar radiation precursors. An example is an extreme UV radiation on 13 January 2015 from the flare which ionized Earth's atmosphere and caused radio blackout in part of the Earth. This hazard happened while our instrument detected two type IV bursts on 12 January 2015 and one on the next day. Affected areas are shown on top of Figure 7.

[Figure]

The intense radio emissions shown in the above figure are associated with major impulsive increases with X-rays and EUV emission. Also energetic particles from the Sun arrive at the Earth within a few hours after the flare and result in a large enhancement of high proton levels which in turn cause shortwave radio blackouts in different parts of the Earth.

Comment#9

**It should also be pointed out in what phase of the solar cycle the observations have been performed in order to evaluate the number, the kind and strength of radio bursts and their connection to solar phenomena.**

Response

The observation was made within the solar cycle 24 in the range October 2014 - September 2015.

---

## Author Comment (AC2)

**Title:** **Space Weather Study through Analysis of Solar Radio Bursts detected by a Single Station CALLISTO Spectrometer.**

Answers to the referee # 2

Firstly, we thank the referee for providing useful comments on our manuscript. Following the referee's comments, we have carefully gone through the manuscript and revised it. **For the sake of convenience, the newly added text in the manuscript is highlighted in boldface.** Herewith, we provide the answers and/or explanations to the referee's comments:

Comment # 1

The paper summarizes the results of the first year of observations with an e-CALLISTO radio spectrograph at Kigali, Rwanda. It gives lists of the observed radio bursts, briefly discusses some statistics, and attempts to draw a few conclusions. The main problem with the paper is that the analysis and interpretation of the events is quite superficial. No qualitatively new results are obtained. I see two options: some more in-depth analysis, in particular putting elaborating on the statistical results and putting them into context of existing work, and/or focusing more on the technical and operational aspect, e.g. by demonstrating why this e-CALLISTO station is a valuable addition to the network. I have addressed particular issues below.

**Comparison to existing event catalogs:**
**In assessing the performance of the instrument, it would be very instructive to compare the number of detected bursts with existing event catalogs. Radio bursts are available from the NOAA Space Weather Prediction Center at least for the second half of 2015 (ftp://ftp.swpc.noaa.gov/pub/indices/events/), so a partial comparison could be done. Such a discussion would significantly strengthen the case of the paper.**

**Response**

Recent studies by Ndacyayisenga et al., 2021 found that out of 12971 type III bursts events reported from the NOAA Space Weather Prediction Centre (SWPC), CALLISTO spectrometers detected 698 type III bursts within the period from 2010 until 2017. Similar analysis by Umuhire et al.,2021 found that among 365 type II bursts, only 107 type II bursts could be detected by e-CALLISTO spectrometers.

The aim of the current paper is not to compare the efficiency of the CALLISTO spectrometer installed at University of Rwanda, College of Education with any other radio spectrometers. The overall objective is to demonstrate the possibility of such a single station of e-CALLISTO network to contribute to the space weather study through observation of solar radio bursts. We appreciate the referee for the suggestion to make comparison of CALLISTO with any other spectrometer. We would think that more emphasis on it might shift the focus of the current paper. However, we now provide sufficient discussion on the space weather part which was initially not done.

CALLISTO instrument was designed in the framework of IHY2007 and the idea of providing a cheap instrument to support developing countries in solar radio astronomy. Total cost for such a telescope was decided to be in the order of US2000$ which every interested institute can afford. This financial constraint was only possible by selecting a cheap antenna (LPDA) with main beam gain of ~6 dB and a low cost frequency agile spectrometer as a back-end. Any high sensitive digital back-end with similar frequency range would cost in the order of 100 times more than a CALLISTO spectrometer.

Comment #2

**Type IV bursts, shocks, and CMEs (line 79 & 114): While the association of type IV bursts with CMEs is high, the claim that they are caused by CME-driven shocks is erroneous. It is well established that type IV bursts are generated by energetic electrons trapped in a magnetic structure, e.g. an erupting flux rope in the framework of a CME. This error has to be corrected.**

**Response**

We thank the referee for pointing out this issue. We agree with him/her that type IV bursts are generated by energetic electrons trapped in a magnetic structure, e.g. an erupting flux rope in the framework of a CME. High association of Type IV bursts with CMEs can only be used to study the kinematics of CMEs and trace out the physical characteristics of the electrons generating type IV bursts. The mistake is corrected.

**Referee #2 Minor issues**:

Comment # 3

**Sect. 2: Technical details of the Callisto system as implemented in Rwanda should be given, including temporal and frequency resolution and antenna characteristics.**

**Response**

Given this low antenna gain of ~6 dB, selected frequency range 45 MHz - 80 MHz, low noise amplifier with ~2 dB noise figure, 300 KHz radiometric spectrometer bandwidth and 1 ms integration leads to a system sensitivity during transit of the Sun in the order of 22 sfu ... 66 sfu. Thus, such a low cost system can never see weak bursts, it can only detect strong bursts during Sun transit ± 3 hours.

A CALLISTO based radio telescope can never detect quiet Sun unless the antenna (LPDA) is replaced by a parabolic dish in the order of at least 5m diameter and an appropriate tracking system. Nevertheless a CALLISTO system allows one to step into solar radio astronomy and study dynamic radio spectra.

Temporal resolution is given by a fixed sampling rate of 800 S/s which is divided into a number of frequency channels, e.g. 200. In this case we get a time resolution of 800 S/s / 200 = 4 spectra per second which corresponds to 250 ms time resolution. For 100 frequency channels we would then get 125 ms time resolution and so forth. Frequency step size is given by the instrument firmware to 62.5 kHz, leading to theoretically 13'200 possible observing frequencies. In practice we select 200 best frequencies out of the maximum observable spectrum 45 MHz- 870 MHz for observation. Best in this context means lowest level of rfi. Different from step-size is the radiometric bandwidth which is given by a ceramic filter of 300 kHz. Detailed specification of the spectrometer can be found here:

> http://www.e-callisto.org/Hardware/eCallistoSpecification.pdf

Detailed specification of the antenna CLP-5130-1 can be found here:

> http://www.cd-corp.com/eng/cma/clp5130.pdf

Comment#4

**Tables: How is the CME onset time defined?**

**Response**

A coronal mass ejection (CME) is a significant release of plasma and accompanying magnetic field from the solar corona. The released plasma can be observed in coronagraph imagery. Its first appearance time in

the coronagraphic field of view  is known as the CME onset time.

Comment #5

**line 82: It is said that a specific type II burst is chosen for its geo-effectiveness. It is more appropriate to say that it was chosen because it was associated with a geo-effective event, since a coronal shock by itself is not geo-effective.**

**Response**

The line is rephrased as follows: Among type II bursts detected by the instrument, we have chosen the type II burst of August 22, 2015 because it is associated with a  geo-effective event.

Comment # 6

**line 97: It is claimed that outward-propagating waves are associated with the bursts. The statement is at odds with the next sentence that associates type III bursts with jets. A jet is not a wave, so please clarify this issue.**

**Response**

The statement is rephrased as follows: With the help of images provided by AIA/SDO, the remaining 11 bursts are associated with leading jets on the west edge.

Comment#7

**line 102: Not clear to me what the authors want to say here - please rephrase.**

**Response**

From Figure 3, it is seen that the jets look the same from 06:55:15 UT to 10:04:05 UT, and it is believed that there is a repetitive type III radio emission associated with them (Chifor et al., 2008, https://doi.org/10.1051/0004-6361:200810265 ).

Comment # 8

**lines 109-112: It should be noted that the latitudinal distribution of type III-associated**

**flares is not surprising, as it just reflects the distribution of flares (and active regions) in general.**

**Response**

The paragraph is modified as follows: Although the small fraction of type III radio bursts, we have plotted the heliographic longitudes and latitudes of the associated solar flares as indicated in Figure 5. I**t is seen that the distribution of flares associated with type III radio bursts originate near the equator (±30◦).** This result is consistent with the findings by Mahender et al. (2020) who found that 125 type III radio bursts among 426 are associated with the solar flares that originated close to the equator (i.e. heliographic latitudes ±23◦).

**Comment #9**

**lines 126 & 127 (and potentially other places: When referring to the instrument please use "spectrometer" instead of spectrogram.**

**Response**

We thank the referee for the suggestion. The mistake is corrected.

---

## Author Comment (AC3)

**Title:** **Space Weather Study through Analysis of Solar Radio Bursts detected by CALLISTO Spectrometer.**

**Answers to the referee #3 comments**

Firstly, we thank the referee for providing useful comments on our manuscript. Following the referee's comments, we have carefully gone through the manuscript and revised it. **For the sake of convenience, the newly added text in the manuscript is highlighted in boldface**. Herewith, we provide the answers and/or explanations to the referee's comments:

(a) **General view**

Comment #1

**The manuscript presents the single station CALLISTO Spectrometer observations in Rwanda. The authors present different types of radio bursts but do not provide even convincing hypothesis on how the radio emission can be employed in the space weather diagnostics. Further, the study presented in this manuscript is very biased regarding the event selection. The sensitivity of CALLISTO instrument is quite low, which indicates that only strong and intense radio bursts will be observed while weak events will not be detected. This effect is enhanced with strong RFI disturbances in the considered observations. As a result, in particular all the type II bursts are very intense events and therefore associated with both flares and CMEs.**

Response:
CALLISTO instrument was designed in the framework of IHY2007 and the idea of providing a cheap instrument to support developing countries through NASA- and UN-initiative in solar radio astronomy. Total cost for such a telescope was decided to be in the order of US2000$ which every interested institute can afford. This financial constraint was only possible by selecting a cheap antenna (LPDA) with a gain of ~6 dB, no tracking system and a low cost frequency agile spectrometer as a back-end.

Given these constraints only strong bursts with several dB above background radiation and rfi can be detected. Any high sensitive digital back-end with similar frequency range would cost in the order of 100 times more than a CALLISTO spectrometer. Nevertheless, such an instrument is very important and adds its contribution to space weather studies especially in developing countries. In fact, the instrument effectively detects strong SRBs and as we know, mostly those strong radio bursts are associated with solar events (CMEs, Solar flares) which are likely to be geoeffective.

Comment # 2

**Unfortunately, it seems that the authors do not fully understand the subject they are addressing. They talk about geoeffectiveness and arrival of the disturbance to Earth, but they do not even mention in situ observations. In the manuscript the question on, how the CME arrival to Earth was estimated and how the associated radio emission helps in that was not addressed. Further, the identification of jets in the presented figure in not correct. Even associations of different types of the radio bursts with eruptive events, such as CMEs and flares is not clear and is some cases incorrect. Figures are also not good, sometimes distorted, but also misinterpreted. Additionally, English language of the manuscript is also quite bad. Due to all mentioned issues, and also the point that the manuscript does not bring any new results, I cannot recommend publication of this manuscript. My suggestion is rejection.**

Response:

The main objective of the study is to show the capability of CALLISTO spectrometers to inform in advance incoming solar phenomena (i.e CME) likely to impact Earth, by for example driving a Geomagnetic storm). This study was limited to indicating correlations between early detected SRBs and subsequent earth impact. The reviewer is correct as authors did not consider using Interplanetary data / measurements for the study. The manuscript has been reviewed to improve to be understood in terms of english.

**Comment #3**

**Unfortunately, I cannot induce the authors to work on the manuscript improvements. I support the idea of the authors to promote their observations, however then the scope of the manuscript needs to be very different. I have provided numerous detailed comments of the manuscript, hoping that will help to the authors to understand what is the problem in the manuscript and how to address it next time.**

Response:

The authors thank you for these useful comments and declare that they like challenging work. Understanding the real problem is a key to the improvement of the current paper. However, we now provide sufficient discussion on the space weather part which was initially not done.

**(b) General comments**

Comment #1

**Employed data set suffers from the preselection effects. The sensitivity of CALLISTO instrument is low, which indicates that only strong and intense radio bursts will be observed while weak events will be not detected. This effect is enhanced with strong RFI disturbances in the considered observations. As a result, all studied radio bursts, all the type II bursts are very intense events. In a case of type II bursts this will result in the 100 % association with both flares and CMEs. The problem of the instrument sensitivity needs to be discussed and data should be compared with observations of some other instrument (other than CALLISTO).**

Response:

We agree with the referee that CALLISTO only observes strong radio bursts. CALLISTO is a frequency-agile spectrometer that is easily transportable and can be used in many observatories. Such a spectrometer is needed to complement interferometers. Its main goal is solar radio observation.

The aim of the current paper is not to compare the efficiency of the CALLISTO spectrometer but to demonstrate the capability of such an instrument which is part of the global network to contribute to the space weather study through observation of solar radio bursts. It is good that the instrument can detect strong radio bursts which in most cases are associated with solar storms that are likely to impact earth's magnetosphere and ionosphere. We appreciate the referee for the suggestion to make a comparison of CALLISTO with any other spectrometer. The instrument used is part of a global network of CALLISTO spectrometers to provide a global coverage. In the current version of the paper, we tried to provide sufficient discussions on the space weather part which was initially not done. We shall also provide a general comparison of the instrument with others.

**Comment # 2**

**The explanation on how the association of studied radio events and CMEs/flares is not present at all. When the authors consider radio emission is associated with eruptive event? What is the time window they use?**

Response:

The current study aims in demonstrating the capability of such an instrument which is part of the

global network to contribute to the space weather study through observation of solar radio bursts. The data used are  provided by CALLISTO station in Rwanda during its first year of operation from October 2014 to September 2015 during the SC 24.  As an advantage, the considered  period overlapped  the availability of multiple spacecraft orbiting the Sun to make an accurate association of the moving radio bursts identified with the occurrence of CMEs.

To confirm the CME -SRBs association, we checked their time coincidence (for this study, CMEs that occurred within one hour of the onset of the radio bursts were considered). we used white–light coronagraph data from the Large Angle and Spectrometic Coronagraph (LASCO C2) onboard the Solar and Heliospheric Observatory (SOHO) and the Sun-Earth Connection Coronal and Heliospheric Investigation (SECCHI) onboard the Solar Terrestrial Relationship Observatory (STEREO) Ahead (A) and Behind (B) spacecrafts. For some cases where the identification was not clear, the CMEs were checked manually around the time of the moving burst events. The CME source regions were checked for events where multiple CMEs occurred at the same time to correlate them with associated bursts using CALLISTO images.

Regarding flare association, the onset, peak and end time of the flare were checked from the x-ray flare plot from the Geostationary Operational Environmental Satellite (GOES). We confirmed the association by checking their time coincidance, most likely when the radio burst is present between the onset and end times of the associated flare, we treat it as flare associated .

**Comment # 3**

**Further, I do not really see how presented study is related to Space Weather.**
**There is no explanation in the manuscript which criterial are used to associate radio bursts**
**and the possible geomagnetic impact.**

Response:

Observed solar Radio bursts (SRBs) do not directly drive geomagnetic storms and other space weather adverses.  However solar radio bursts  are observed often associated with flares and/or CMEs which are precursors and major drivers of space weather manifestations including geomagnetic storms. As such, SRBs are used as a proxy and part of good data for model prediction of space weather. In the new version of the paper sufficient discussions  with linkage to space weather are provided which was initially not done.

**Comments # 4**

**Number of definitions is incorrect, e.g. association of the type IV burst and the sock wave...
Some figures are not well done and some are incorrectly Interpreted.**

Response:

We thank the referee for pointing out this issue. The part of type IV bursts and their association with eruptive events is revised and corrected for the mistakes. The figures are now well interpreted.

(c) **Detailed comments**

**Comment # 1**

**Page 1, line 20: Abbreviation "CMEs" needs to be introduced.**

Response:

The line is modified as :  …… often associated with coronal mass ejections (CMEs) and ……...

Comment # 2

**Page 2, line 25: The authors write: "they are generated when solar flares send electron beams streaming into the heliosphere via plasma mechanism".
This statement is not completely correct, the electron beams generated during the flaring process propagate along open or quasi-open magnetic field lines and they can induce Langmuir waves which will the cause the radio emission at plasma frequency and/or its harmonic.**

**Response:**

We thank the referee.  The paragraph is modified.  …. they are generated by electrons propagating along open magnetic field lines and trigger the plasma oscillations (also known as Languir waves) during their travel in the solar corona and interplanetary medium (IPM) (Ginzburg and Zhelezniakov, 1958;  Zheleznyakov, 1970, etc).

Comment # 3

**Page 2, line 26: The authors write that type III bursts duration is 1-3 s. However, as duration of the type IIIs depends on the observing frequency it is necessary to specify the frequency range together with the durations.**

Response:
The statement is formulated as follows: Type III radio bursts drift from ~ 200 MHz to 30 MHz in less than 10 s and reach 30 kHz in about 1 h.

Comment # 4

**Page 2, line 27: The authors say: "The impulsive flares in X-ray and/or H α frequencies..." It should be x-ray and $H\alpha$ wavelengths.**

Response:

We thank the referee. The error is corrected.

Comment # 5

**Page 2, line 31: The authors already used the abbreviation "CME" so no need to use the full name.**

Response:

The abbreviation is maintained.

Comment # 6

**Page 2, line 36: The authors should also mention very important and recently refurbished Nançay observations.**

Response:

We have added the recommended observatory.

Comment # 7

**Page 2 footnoote 3. The link for the "callistoQuicklooks" does not work.**

Response:
We have provided a correct link. http://soleil.i4ds.ch/solarradio/callistoQuicklooks/

Comment # 8

**Page 3, line 58: What does mean 'through a channel of 80.9 MHz to 45 MHz?**
 Response:
The statement is rephrased as follows:  All bursts were observed in the frequency range of 45 MHz to 80.9 MHz.

Comment  # 9

**Table 1: The authors do not define when they consider the radio burst to be "associated with flares and/or CMEs". What is the time window in which two phenomena are considered to be related?**

Response:

To confirm the CME -SRBs association, we checked their time coincidence (for this study, CMEs that occurred within one hour of the onset of the radio bursts were considered). we used white–light coronagraph data from the Large Angle and Spectrometric Coronagraph (LASCO C2) onboard the Solar and Heliospheric Observatory (SOHO) and the Sun-Earth Connection Coronal and Heliospheric Investigation (SECCHI) onboard the Solar Terrestrial Relationship Observatory (STEREO) Ahead (A) and Behind (B) spacecrafts. For some cases where the identification was not clear, the CMEs were checked manually around the time of the moving burst events. For the events where multiple CMEs occurred at the same times, we checked the source region of the moving bursts from CALLISTO images to associate them with the correct CME.

Regarding flare association, the onset, peak and end time of the flare were checked from the x-ray flare plot from the Geostationary Operational Environmental Satellite (GOES). We confirmed the association by checking their time coincidance, most likely when the radio burst is present between the onset and end times of the associated flare, we treat it as flare associated .

Refer to comment # 2

Comment # 10
**Page 4, line 75: The authors state wrongly: "type IV bursts may be due to small-scale feature events present in the solar corona. First, the type IV bursts are generally associated with CMEs, so they cannot be put in the same category as type III bursts that are not flare associated. Second, what are 'small-scale feature events' ? Do the authors refer to nanoflares, small scale reconnection events or something else?**

Response:

We thank the referee for this comment. Type III radio bursts and type IVs are separate, no way to put them in the same category. The two lines are modified as follows: The remaining non-flare associated type III radio bursts may be due to small-scale feature events present in the solar corona.

Comment # 11

**Page 4, line 78: The authors write incorrect, type II bursts and not type IV radio bursts are triggered by CME-driven shocks. The type IV emission is considered to be associated with CMEs, but not CME-driven shocks.**

**Response:**

The statement is rephrased as follows:  In similar way, it is believed that the majority (~82%) of type IV radio bursts are associated with CMEs.

Comment # 12

**Page 4, line 79: On figure 2 there is absolutely no information that would indicate that studied event was geoeffective. So, the following statement has no ground: "the one of August 22, 2015 is chosen based on its geoeffectiveness as displayed in Figure 2."**

Response:

The statement is rephrased as follows: Among type II bursts detected by the instrument, the one of August 22, 2015 is chosen as due to its visible dynamic spectrum, hence flexible for spectral data analysis.

Comment # 13
**Figure 2 (Page 7): Panel c) does not show the CME itself but it shows the associated EUV wave, one of the on disc signatures of CME. Considering blue circle as a CME-driven shock is very provisional consideration, and needs to be better justified and with some references.**

Response:

In Figure 2 c, a LASCO C2 CME image will be used to clear the signature of the CME appearance.
The CME shock height was taken as the radius of the circle fitted to the outermost part of the disturbance; the assumption is  that the disturbance expands spherically above the solar

surface. Hence the CME shock height is shown by the red line. More details on the method used can be found in Gopalswamy et al., 2013.

Comments # 14

**Page 4, line 81: What means "The event has a band split fundamental structure with the corresponding frequencies ranging between 46-56 MHz and 46-75 MHz,"? I believe it should be written something like: the fundamental band of the type II burst shows the band split. It should be also explained what is band split, and what are the listed frequencies? References need to be added describing the band split, eg. Vrsnak et al., 2001 and some recent work like Mahrous et al., 2018.**

Response:

We thank the referee for the useful comment and suggestion. The detailed discussions are provided in the new version of the manuscript.

Comment # 15

**Page 4, line 84: The authors state that the CME occurred at 07:12 UT. Where did he occurred, in the EUV images or in coronagraph images, and which one SOHO/LASCO C2? All this needs to be specified.**

Response:
The sentence is revised and rephrased. The August 22, 2015 CME shock was only detected in SDO/AIA as a EUV wave as shown in Figure 2(c). However, the CME evolution could be observed and measured within LASCO C2 from 07:12 UT after the shock.

Comment # 16

**Page 4, line 85-86: I am wondering how the CME with the speed of about 640 km/s arrived from the Sun to Earth in less than 24 hours?! This is surely not possible. The authors should check the association of the solar event and the one observed in situ. The disturbance propagating with the speed of about 700 km/s will need about 2.5 days to come to Earth. Further, the way shock height is estimated is not scientifically justified.**

Response:
We thank the referee for this comment.
Depending on the speed, a typical CME reaches the Earth between 1- 5 days.

We report that a CME of 2015/08/22 at 07:12 UT with a speed of 643 Km/s reached Earth on 2015/08/24 causing a minor storm (~ -30 nT).

The CME shock height was taken as the radius of the circle fitted to the outermost part of the disturbance; the assumption is that the disturbance expands spherically above the solar surface. Hence the CME shock height is shown by the red line. More details on the method used can be found in Gopalswamy et al., 2013.

Comment # 17

**Page 5, line 88: What does mean that flare occurred at 06:49 UT? Was that the flare start of the maximum? All this needs to be specified.**

Response:

The statement is modified as follows: ….., that started at 06:49 UT and …….

Comment # 18

**Page 5, line 89-90: The simultaneous occurrence of the flare and CME is not the point which makes difficult understanding which one of them is generating coronal shock. It is the simultaneous flare impulsive phase and the CME acceleration phase that does not allow us to clearly understand if the shock is flare-generated or CME-driven.**

Response:

We thank the referee for this important comment. The sentence is revised .
.

Comment # 19

**Figure 3 (page 8): In the presented images no clear jets are observed. What the authors call bright 'nods' are the brightenings observed when the closed loop system moves. Usually, jets are observed propagating along the open filed lines, and I see nothing like this in the presented figure. And in particular, that should be the case when jets are associated with type III radio bursts – signatures of fast electron beam propagating along open field lines. We can observe open field lines in the figure but none of them shows brightening indicating jet propagation.**

Response:

The aim of Figure 3 is to show the potential site where non-flare type III radio bursts are believed to originate. We have looked at the Sun and taken portions of the Sun containing the region of interest. We will increase the size of the images but we expected that the expanded edge corresponds to a jet. Hence if it shows the region of open magnetic field lines, that is the path of electron beams emitting type III radio bursts.

Comment # 20

**Page 5, line 102-103: This sentence has no sense. Where the authors see propagating waves?**

Response:
We thank the referee.  we have removed the sentence.

Comment # 21

**Page 5, line 104-105: This sentence is not justified. In order to associate type III bursts with CME at least the source positions of type III bursts need to be checked and compared with the source region of CMEs. The authors should study literature a bit better. Namely, numerous type III bursts can appear during the so called type III storms that can last for days. And, it is generally considered that they are associated with the complex active regions observed on the visible side of the solar disc.**

Response:

We thank the referee. We used the CME catalog and images at https://helioviewer.org to confirm the association between type III bursts and CMEs. Since we did not check their source position, we have dropped the statement.

Comment # 22

**Page 5, line 113: What does this sentence mean?**

Response: We thank the referee. The sentence was removed as it has no meaning.

Comment # 23

**Page 5, line 114: What is 'backbone of the CME-driven shocks' ? Why the authors think type IV bursts are 'poorly associated with flares? This statement needs to be justified and references provided.**

Response:

It is well established that type IV bursts are generated by energetic electrons trapped in a magnetic structure, e.g. an erupting flux rope in the framework of a CME.
Type IV radio bursts are classified into two categories: Stationary type IV bursts and moving type IV bursts. Stationary type IVs prevail over the presence of non-thermal electrons in the solar corona for several hours in relationship with flare and the liftoff of a CME. Moving type IV radio bursts are believed to originate due to synchrotron or gyro-synchrotron emitting electrons, gyrating inside helical magnetic fields within the CME flux rope.
Thus, it is erroneous to conclude that type IV bursts originate from the backbone of the CME-driven shocks and that they are poorly associated with solar flares.
Therefore, the statement is rephrased and modified as: ***Table 3 lists all type IV radio bursts observed by the network and their association with the solar phenomena. From this table, it can be seen that type IV bursts are highly associated with CMEs which indicates that their presence can be used to map the location of trapped electrons and studying the CME kinematics during the phases of eruption processes (Kumari et al., 2021).***

N.B: Table 2 will be removed (Referee # 1)

Comment # 24

**Page 6, line 117 – 118: Figure 6 does not show any type IV radio bursts, but it shows somehow distorted images of CMEs in the SOHO/LASCO C2 filed of view. The running difference images of SDO, 171 A are also very bad. It is not completely clear where is the solar limb and the Sun seems distorted.**

Response:
Like for other radio bursts, the association of type IV bursts with CMEs were carried out using the catalog from http://hec.helio-vo.eu/hec/hec_gui.php
From this source, the catalog comprised SOHO/LASCO and STEREO A/B. But for some cases we used https://helioviewer.org (currently not opening) to see whether there was a CME.
Figure 6 aims to show the images of CMEs and their corresponding potential site on the solar surface for the dates where non-flare associated type IV radio bursts were detected. The figure does not indicate any type IV radio burst at all.

We now modify the figure as per attached below with the aim of showing the magnetic loops embedded in the solar corona  where we expect to be the site of trapped electrons responsible for Type IV emissions. The figure will display a sample of six AIA/SDO images in 171 Å band filter corresponding to the dates of type IV radio bursts in the January 2015.

[Figure]

[Figure]

**Comment # 25**

**Page 6, line 118-119: How is this visible?**

Response:
The initial statement is: ***The first two type IV bursts are accompanied by CMEs followed by another two without CMEs and then two type IV with CMEs.***
We have dropped the statement as an effect of Figure 6 reformulation.

Comment # 26
**Page 6, line 122-124: The authors first state the some type IV radio burst may be not associated with flares and CMEs. And in the following sentence they state that this kind of type IV bursts coincide with the decaying phase of flares or post eruption loops. These two sentences are in contradiction.**

Response:
Page 6, lines 122 - 124 are modified as follows:

*It is trustworthy that type IV radio bursts coincide with the decaying phase of flares and/or triggered by post-eruption loops (Morosan et al., 2019; Kumari et al., 2021) but they may lack both association with flares and CME eruptions (Table 3).*

Comment # 27

**Page 6, line 124: This is very general sentence, not related at all to presented study. It cannot be considered as conclusion.**

Response:

We thank the referee for this comment. The statement is dropped.

Comment #28

**Page 7, line: By e-CALLISTO we can be continuously monitor radio emission and not Space Weather.**

Response:

We thank the referee. The statement is corrected.

Comment # 29

**Page 9, line 141: Which radio bursts may be used as a precursor for space weather diagnostics? And how? The authors did not really provide convincing evidence, actually any evidence how we can use radio observations for space weather diagnostics.**

Response:

The line and the paragraph is modified as follows:

Observed radio bursts (type II, III and type IV) may be used as a precursor for space weather diagnostics since they are often associated with space weather drivers such as CMEs and SFs. Solar radio bursts can be detected at the ground level, hence serving as the advance warning of incoming associated solar transient events. Therefore, they provide insights to predict/forecast the incoming space weather hazards. HF communications and the ionosphere are disturbed by the X-ray and EUV wavelengths along with the solar energetic particles that reach Earth whinin few hours and they are signatures of solar flares and CMEs.

---

## Author Response (AR2)

**Space Weather Study through Analysis of Solar Radio Bursts detected by a Single Station CALLISTO Spectrometer**

T. Ndacyayisenga, A. C. Umuhire, J. Uwamahoro, C. Monstein

August 16, 2021

Firstly, we thank the referee for providing useful comments on our manuscript. Following the referee's comments, we have carefully gone through the manuscript and revised it. **For the sake of convenience, the newly added text in the manuscript is highlighted in boldface**. Herewith, we provide the answers and/or explanations to the referee's comments and the text in blue highlights the changes made in the manuscript.

**1 Answers to the referee #1 comments:**

1. **The paper is a summary of results of solar radio bursts analysis observed by the e-Compound Astronomical Low cost Low-frequency Instrument for Spectroscopy and Transportable Observatory (e-CALLISTO), comprising the period Oct 2014 – Sept 2015, at the University of Rwanda, College of Education. Main motivation of this study is formulated to use radio burst data to check their correlation with solar activities**.

   **Answer:** We thank the referee for his/her suggestion. The objective is reformulated.

2. **In Chapter 1 – "Introduction" a (too) short overview is presented on the morphologies of the various types of solar radio bursts, indicating the registration of 5 Type II bursts, 175 Type III bursts, and 22 Type IV bursts within the given period of observation. The data base of Type II and Type IV bursts is rather scarce, apparently owed to the limited sensitivity of the used antennas. At this point a more detailed description of the applied instruments (antennas and backend facilities) is strongly recommended.**

   **Answer:**

   (a) The introduction has been revised. The review of various types of solar radio bursts is limited and is beyond the scope of the current article. We have provided references in the literature.
   The standard situation is that type III radio bursts are most intense and frequent emission from the Sun than type II and type IV bursts, all depend upon the solar activities. The number of events reported herein has no relationship with brief

introduction given. For example in Ndacyayisenga et al., 2021, they reported a total of 12971 type III radio bursts from Space Weather Prediction Centre (SWPC) occurred between 2010 and 2017, while in the same range, only 426 type II radio bursts have occurred.

(b) CALLISTO instrument was designed in the framework of IHY2007 and the idea of providing a cheap instrument to support developing countries in solar radio astronomy. Total cost for such a telescope was decided to be in the order of US2000$ which every interested institute can afford. This financial constrain was only possible by selecting a cheap antenna (LPDA) with main beam gain of ∼6 dB and a low cost frequency agile spectrometer as a back-end. Any high sensitive digital back-end with similar frequency range would cost in the order of 100 times more than a CALLISTO spectrometer. Given this low antenna gain, selected frequency range 45 MHz - 80 MHz, low noise amplifier with ∼2 dB noise figure, 300 KHz radiometric spectrometer bandwidth and 1 ms integration leads to a system sensitivity during transit of the Sun in the order of 22 sfu to 66 sfu. Thus, such a low cost system can never see weak bursts, it can only detect strong bursts during Sun transit +/- 3 hours. A CALLISTO based radio telescope can never detect quiet Sun unless the antenna (LPDA) is replaced by a parabolic dish in the order of at least 5m diameter and an appropriate tracking system. Nevertheless a CALLISTO system allows to step into solar radio astronomy and studying dynamic radio spectra.

(c) However, in section 2 we briefly discussed the intrument specifications including the antenna with subsequent comparison with other spectrometers.

3. **Among the mentioned spaceborne and ground-based radio observatories (lines 34 − 39) reference should be given to renowned radio telescopes of Nancay (France) and Charkiv and Poltava (Ukraine), resp., which provide with their huge arrays excellent observations in the decameter range**.

   **Answer:** We thank the referee for the suggestion. The recommended instruments have been added to the appropriate list. Page 3, lines 76 − 77.

4. **a Chapter 2 − "Observation and methods" a comparison and supplimentary investigation of the e-CALLISTO results with observations of other stations would enable a better classification and judgement of the quality of the e-CALLISTO results obtained at the College of Education, University of Rwanda. Among other stations the use of the world-wide distributed e-CALLISTO network just offers this opportunity**.

   **Answer:**

   (a) The aim of the current study is to demonstrate the possibility of a single station to the contribution of the space weather studies using observations of solar radio bursts. To test the efficiency of the CALLISTO with any other spectrometers may shift the purpose of this paper.
   Using the e-CALLISTO data, it is reported in Ndacyayisenga et al., 2021 that the e-CALLISTO observed 698 intense type III radio bursts among 12971 bursts from Space Weather Prediction Center (SWPC) within the range of 2010 - 2017.

In the similar range only 426 type II radio bursts are on SWPC. Similar analysis by Umuhire et al.,2021 found that among 365 type II bursts, only 107 type II bursts could be detected by e-CALLISTO spectrometers.

(b) This section has been divided into two sections namely observations and methods. section 2 (observations) describes the instrument and subsequent comparison with other spectrometers. Section 3 (methods) details the methods used to process the data for an analysis.

5. **In Chapter 3 – "Results and discussions" the obtained radio data are listed in tables (including an online-link for the Type III burst data which however – checked on different days - is unavailable!) with the "associated" solar phenomena (Table 1 and Table 3). In this respect the terms "associated" and "related" (line 72) need a clear definition. Table 2 is trivial and its content can be described in not more than two short sentences.**

   **Answer:** We thank the referee for pointing out these issues.
   We have included in the text a correct link which is also given here: `http://www.e-callisto.org/GeneralDocuments/Type_III_radio_bursts_2014_2015_kigali.pdf`
   The term 'related' is removed and keep 'associated' which has a clear meaning in the context. However, table 2 was removed. Page 7, line 137.

6. **The solar radio burst observed on Aug 22, 2015, had a frequency stop at 45 MHz (according to Table 1) whereas in the text (line 82) 46 MHz are given.**

   **Answer:**

   (a) The difference between the two values arose from typing. The correct value is 45 MHz and we have corrected the text accordingly.

   (b) This radio burst is included in Table 1 but its discussion was removed in the text because it was described in another paper, see `http://doi.org/10.1016/j.asr.2021.06.029`.

7. **Figure 2c visualizes a CME image where the red line is considered as the CME shock height. This needs a clear physical justification.**

   **Answer:**

   (a) The CME shock height was taken as the radius of the circle fitted to the outermost part of the disturbance; the assumption is that the disturbance expands spherically above the solar surface. Hence the CME shock height is shown by the red line. More details on the method used can be found in Gopalswamy et al., 2013.

   (b) This figure was removed.

8. **Figure 3 shows running difference images with the last image taken at 10:04:05 UT, correctly cited in the figure legend, whereas in line 103 the time 10:05 UT is mentioned.**

   **Answer:** We thank the referee for finding it out. The mistake is corrected.

9. **As the title of the paper indicates ("Space Weather Study ...") this branch of space physics comprises the varying conditions within the solar system, caused by the sun, including near-Earth conditions, e.g. in the magnetosphere and ionosphere. At least some examples of strong solar radio emission observed by e-CALLISTO and connected with solar plasma phenomena (flares, CMEs, shock waves) and their impact on Earth would clearly demonstrate the "main motivation of this study" as mentioned above.**

   **Answer:**

   (a) The authors thank the referee for raising this issue. It is clear that this part was missed out which led to the title to be a bit irrelevant.
   We have taken care of this section and included a discussion with supporting examples. To this end using space weather website, it is seen that the flare associated with type II bursts given in Table 1 have given rise to an alert of incoming aurora phenomena (aurorae) in polar regions. Type IV bursts are attributed to electrons trapped in closed field lines in the post-flare arcades and have high degree association with solar energetic particle events (White, 2007). Thus they can be used as extreme solar radiation precursors. An example is an extreme UV radiation on 13 January 2015 from the flare which ionized Earth's atmosphere and caused radio blackout in part of the Earth.

   (b) The section 4 of results and discussions is composed by two subsections: 4.1 Statistical analysis of observed radio bursts and 4.2 Space weather monitoring. Under space weather monitoring, we attempted to show the implications of solar radio bursts on space weather.

10. **It should also be pointed out in what phase of solar cycle the observations have been performed in order to evaluate the number, the kind and strength of radio bursts and their connection to solar phenomena.**

    **Answer:** The observation was made within the solar cycle 24 in the range of October 2014 - September 2015. [Page 4, line 112]

**2 Answers to the referee # 2 comments:**

The paper summarizes the results of the first year of observations with an e-CALLISTO radiospectrograph at Kigali, Rwanda. It gives lists of the obsetrved radio bursts, briefly discusses some statistics, and attempts to draw a few concluions. The main problem with the paper is that the analysis and interpretation of the events is quite superficial. No qualitatively new results are obtained. I see two options: some more in-depth analysis, in particular putting elaborating on the statistical results and putting them into context of existing work, and/or focusing more on the technical and operational aspect, e.g. by demonstrating why this e-CALLISTO station is a valuable addition to the network. I have addressed particular issues below.

1. **Comparison to existing event catalogs:**
   **In assessing the performance of the instrument, it would be very instructive**

to compare the number of detected bursts with existing event catalogs. Radio bursts are available from the NOAA Space Weather Prediction Center at least for the second half of 2015 (ftp://ftp.swpc.noaa.gov/pub/indices/events/), so a partial comparison could be done. Such a discussion would significantly strengthen the case of the paper.

**Answer:**

(a) Recent studies by Ndacyayisenga et al., 2021 found that out of 12971 type III bursts events reported from the NOAA Space Weather Prediction Centre (SWPC), CALLISTO spectrometers detected 698 type III bursts within the period from 2010 until 2017. Similar analysis by Umuhire et al.,2021 found that among 365 type II bursts, only 107 type II bursts could be detected by e-CALLISTO spectrometers.

(b) The aim of the current paper is not to compare the efficiency of the CALLISTO spectrometer installed at University of Rwanda, College of Education with any other radio spectrometers. The overall objective is to demonstrate the possibility of such a single station of e-CALLISTO network to contribute to the space weather study through observation of solar radio bursts. We appreciate the referee for the suggestion to make comparison of CALLISTO with any other spectrometer. We would think that more emphasis on it might shift the focus of the current paper. However, we now provide sufficient discussion on the space weather part which was initially not done.

(c) CALLISTO instrument was designed in the framework of IHY2007 and the idea of providing a cheap instrument to support developing countries in solar radio astronomy. Total cost for such a telescope was decided to be in the order of USD 2000 which every interested institute can afford. This financial constraint was only possible by selecting a cheap antenna (LPDA) with main beam gain of $\sim 6$ dB and a low cost frequency agile spectrometer as a back-end. Any high sensitive digital back-end with similar frequency range would cost in the order of 100 times more than a CALLISTO spectrometer.

(d) Inital section 2 (Observations and Methods) is now divided into two sections: observations and methods. section 2 (observations) describes the instrument and subsequent comparison with other spectrometers. Section 3 (methods) details the methods used to process the data for an analysis.

2. **Type IV bursts, shocks, and CMEs (line 79 & 114): While the association of type IV bursts with CMEs is high, the claim that they are caused by CME-driven shocks is erroneous. It is well established that type IV bursts are generated by energetic electrons trapped in a magnetic structure, e.g. an erupting flux rope in the framework of a CME. This error has to be corrected.**

**Answer:** We thank the referee for pointing out this issue. We agree with him/her that type IV bursts are generated by energetic electrons trapped in a magnetic structure, e.g. an erupting flux rope in the framework of a CME. High association of Type IV bursts with CMEs can only be used to study the kinematics of CMEs and trace out

the physical characteristics of the electrons generating type IV bursts. The mistake is corrected.

3. **The conclusions are quite weak. The fact that all type II bursts were flare-associated does not add any useful information on their origin, given that they were all associated to CMEs as well**.

   **Answer:** These conclusions have been revised throughout the text.

**2.1 Referee #2 Minor issues:**

1. **Sect. 2: Technical details of the Callisto system as implemented in Rwanda should be given, including temporal and frequency resolution and antenna characteristics**.

   **Answer:**

   (a) Given this low antenna gain of $\sim 6$ dB, selected frequency range 45 MHz – 80 MHz, low noise amplifier with $\sim 2$ dB noise figure, 300 KHz radiometric spectrometer bandwidth and 1 ms integration leads to a system sensitivity during transit of the Sun in the order of 22 sfu ... 66 sfu. Thus, such a low cost system can never see weak bursts, it can only detect strong bursts during Sun transit $\pm 3$ hours.

   (b) A CALLISTO based radio telescope can never detect quiet Sun unless the antenna (LPDA) is replaced by a parabolic dish in the order of at least 5m diameter and an appropriate tracking system. Nevertheless a CALLISTO system allows one to step into solar radio astronomy and study dynamic radio spectra.

   (c) Temporal resolution is given by a fixed sampling rate of 800 S/s which is divided into a number of frequency channels, e.g. 200. In this case we get a time resolution of 800 S/s / 200 = 4 spectra per second which corresponds to 250 ms time resolution. For 100 frequency channels we would then get 125 ms time resolution and so forth. Frequency step size is given by the instrument firmware to 62.5 kHz, leading to theoretically 13'200 possible observing frequencies. In practice we select 200 best frequencies out of the maximum observable spectrum 45 MHz – 870 MHz for observation. Best in this context means lowest level of rfi. Different from step-size is the radiometric bandwidth which is given by a ceramic filter of 300 kHz. Detailed specification of the spectrometer can be found here: `http://www.e-callisto.org/Hardware/eCallistoSpecification.pdf` and Detailed specification of the antenna CLP-5130-1 can be found here: `http://www.cd-corp.com/eng/cma/clp5130.pdf`

   (d) Inital section 2 (Observations and Methods) is now divided into two sections: observations and methods. section 2 (observations) describes the instrument and subsequent comparison with other spectrometers. Section 3 (methods) details the method used to process the data for an analysis.

2. **Tables: How is the CME onset time defined?**

   **Answer:** A coronal mass ejection (CME) is a significant release of plasma and accompanying magnetic field from the solar corona . The released plasma can be observed

in coronagraph imagery. Its first appearance time in the coronagraphic field of view is known as the CME onset time.

3. **line 82: It is said that a specific type II burst is chosen for its geo-effectiveness. It is more appropriate to say that it was chosen because it was associated to a geo-effective event, since a coronal shock by itself is not geo-effective.**

   **Answer:**
   The statement was removed together with the event because its linked figure was described in another paper, see http://doi.org/10.1016/j.asr.2021.06.029.

4. **line 97: It is claimed that outward-propagating waves are associated with the bursts. The statement is at odds with the next sentence that associates type III bursts with jets. A jet is not a wave, so please clarify this issue.**

   **Answer:** The statement is rephrased as follows. With the help of images provided by AIA/SDO, a region of open magnetic field lines as the signature of the electron stream responsible for type III radio bursts is indicated. Page 8, lines 170 – 171.

5. **line 102: Not clear to me what the authors want to say here - please rephrase.**

   **Answer:** The statement is rephrased as follows. From Figure 4, it is seen that the position of the open magnetic field lines is stationary from 06:55:15 UT to 10:04:05 UT, and hence one can believe that there is a repetitive type III radio emission originating from the same source (e.g., Chifor et al., 2008). Page 8, line 176 – 177.

6. **lines 109-112: It should be noted that the latitudinal distribution of type III-associated flares is not surprising, as it just reflects the distribution of flares (and active regions) in general.**

   **Answer:** Although the small fraction of type III radio bursts, we have plotted the heliographic longitudes and latitudes of the associated solar flares as indicated in Figure 6. **It is seen that the distribution of flares associated with type III radio bursts originate near the equator** ($\pm 30°$). Page 8, lines 186 – 187.

7. **lines 126 & 127 (and potentially other places: When referring to the instrument please use "spectrometer" instead of spectrogram.**

   **Answer:** We thank the referee for the suggestion. The mistake is corrected.

**3 Answers to the referee # 3 Comments**

**3.1 Referee # 3 General view**

1. **The manuscript presents the single station CALLSTO Spectrometer observations in Rwanda. The authors present different types of radio bursts but do not provide even convincing hypothesis how the radio emission can be employed in the space weather diagnostics. Further, the study presented**

in this manuscript is very biased regarding the event selection. The sensitivity of CALLISTO instrument is quite low, which indicates that only strong and intense radio bursts will be observed while weak events will be not detected. This effect is enhanced with strong RFI disturbances in the considered observations. As a result, in particular all the type II bursts are very intense events and therefore associated with both flares and CMEs.

**Answer:** CALLISTO instrument was designed in the framework of IHY2007 and the idea of providing a cheap instrument to support developing countries through NASA- and UN-initiative in solar radio astronomy. Total cost for such a telescope was decided to be in the order of USD 2000 which every interested institute can afford. This financial constraint was only possible by selecting a cheap antenna (LPDA) with a gain of $\sim 6$ dB, no tracking system and a low cost frequency agile spectrometer as a back-end.

Given these constraints only strong bursts with several dB above background radiation and rfi can be detected. Any high sensitive digital back-end with similar frequency range would cost in the order of 100 times more than a CALLISTO spectrometer. Nevertheless, such an instrument is very important and adds its contribution to space weather studies especially in developing countries. In fact, the instrument effectively detects strong SRBs and as we know, mostly those strong radio bursts are associated with solar events (CMEs, Solar flares) which are likely to be geoeffective.

2. **Unfortunately, it seems that the authors do not fully understand the subject they are addressing. They talk about geoeffectiveness and arrival of the disturbance to Earth, but they do not even mention in situ observations. In the manuscript the question on, how the CME arrival to Earth was estimated and how the associated radio emission helps in that was not addressed. Further, the identification of jets in the presented figure in not correct. Even associations of different types of the radio bursts with eruptive events, such as CMEs and flares is not clear and is some cases incorrect. Figures are also not good, sometimes distorted, but also misinterpreted. Additionally, English language of the manuscript is also quite bad. Due to all mentioned issues, and also the point that the manuscript does not bring any new results, I cannot recommend publication of this manuscript. My suggestion is rejection.**

**Answer:** The main objective of the study is to show the capability of CALLISTO spectrometers to inform in advance incoming solar phenomena (i.e CME) likely to impact Earth, by for example driving a Geomagnetic storm). This study was limited to indicating correlations between early detected SRBs and subsequent earth impact. The reviewer is correct as authors did not consider using Interplanetary data / measurements for the study. The manuscript has been reviewed to improve to be understood in terms of english.

3. **Unfortunately, I cannot induce the authors to work on the manuscript improvements. I support the idea of the authors to promote their observations, however then the scope of the manuscript needs to be very different. I have provided numerous detailed comments of the manuscript, hoping**

that will help to the authors to understand what is the problem in the manuscript and how to address it next time.

**Answer:** The authors thank you for these useful comments and declare that they like challenging work. Understanding the real problem is a key to the improvement of the current paper. However, we now provide sufficient discussion on the space weather part which was initially not done.

**3.2  Referee # 3 General comments**

1. **Employed data set suffers from the preselection effects. The sensitivity of CALLISTO instrument is low, which indicates that only strong and intense radio bursts will be observed while weak events will be not detected. This effect is enhanced with strong RFI disturbances in the considered observations. As a result, all studied radio bursts, all the type II bursts are very intense events. In a case of type II bursts this will result in the 100 % association with both flares and CMEs. The problem of the instrument sensitivity needs to be discussed and data should be compared with observations of some other instrument (other than CALLISTO).**

    **Answer:**

    (a) We agree with the referee that CALLISTO only observes strong radio bursts. CALLISTO is a frequency-agile spectrometer that is easily transportable and can be used in many observatories. Such a spectrometer is needed to complement interferometers. Its main goal is solar radio observation.

    (b) The aim of the current paper is not to compare the efficiency of the CALLISTO spectrometer but to demonstrate the capability of such an instrument which is part of the global network to contribute to the space weather study through observation of solar radio bursts. It is good that the instrument can detect strong radio bursts which in most cases are associated with solar storms that are likely to impact earth's magnetosphere and ionosphere. We appreciate the referee for the suggestion to make a comparison of CALLISTO with any other spectrometer. The instrument used is part of a global network of CALLISTO spectrometers to provide a global coverage. In the current version of the paper, we tried to provide sufficient discussions on the space weather part which was initially not done. We shall also provide a general comparison of the instrument with others.

    (c) Inital section 2 (Observations and Methods) is now divided into two sections: observations and methods. section 2 (observations) describes the instrument and subsequent comparison with other spectrometers. Section 3 (methods) details the methods used to process the data for an analysis.

2. **The explanation on how the association of studied radio events and CMEs/flares is not present at all. When the authors consider radio emission is associate with eruptive event? What is the time window they use?**

    **Answer:**

(a) The current study aims in demonstrating the capability of such an instrument which is part of the global network to contribute to the space weather study through observation of solar radio bursts. The data used are provided by CAL-LISTO station in Rwanda during its first year of operation from October 2014 to September 2015 during the SC 24. As an advantage, the considered period overlapped the availability of multiple spacecraft orbiting the Sun to make an accurate association of the moving radio bursts identified with the occurrence of CMEs.

(b) To confirm the CME -SRBs association, we checked their time coincidence (for this study, CMEs that occurred within one hour of the onset of the radio bursts were considered). we used white-light coronagraph data from the Large Angle and Spectrometric Coronagraph (LASCO C2) onboard the Solar and Heliospheric Observatory (SOHO) and the Sun-Earth Connection Coronal and Heliospheric Investigation (SECCHI) onboard the Solar Terrestrial Relationship Observatory (STEREO) Ahead (A) and Behind (B) spacecrafts. For some cases where the identification was not clear, the CMEs were checked manually around the time of the moving burst events. The CME source regions were checked for events where multiple CMEs occurred at the same time to correlate them with associated bursts using CALLISTO images.

(c) Regarding flare association, the onset, peak and end time of the flare were checked from the X-ray flare plot from the Geostationary Operational Environmental Satellite (GOES). We confirmed the association by checking their time coincidance, most likely when the radio burst is present between the onset and end times of the associated flare, we treat it as flare associated.

3. **Further, I do not really see how presented study is related to Space Weather. There is no explanation in the manuscript which criterial are used to associate radio bursts and the possible geomagnetic impact.**

   **Answer:**

   (a) Observed solar Radio bursts (SRBs) do not directly drive geomagnetic storms and other space weather adverses. However solar radio bursts are observed often associated with flares and/or CMEs which are precursors and major drivers of space weather manifestations including geomagnetic storms. As such, SRBs are used as a proxy and part of good data for model prediction of space weather. In the new version of the paper sufficient discussions with linkage to space weather are provided which was initially not done.

   (b) The section 4 of results and discussions is composed by two subsections: 4.1 Statistical analysis of observed radio bursts and 4.2 Space weather monitoring. Under space weather monitoring, we attempted to show the implications of solar radio bursts on space weather.

4. **Number of definitions is incorrect, e.g. association of the type IV burst and the sock wave... Some figures are not well done and some are incorrectly interpreted.**

**Answer:** We thank the referee for pointing out this issue. The part of type IV bursts and their association with eruptive events is revised and corrected for the mistakes. The figures are now well interpreted.

**3.3 Referee # 3 Detailed comments**

1. **Page 1, line 20: Abbreviation "CMEs" needs to be introduced.**

   **Answer:** The solar space weather events like Coronal Mass Ejections (CMEs) and solar flares .... Page 1, line 16.

2. **Page 2, line 25: The authors write: "they are generated when solar flares send electron beams streaming into the heliosphere via plasma mechanism". This statement is not completely correct, the electron beams generated during the flaring process propagate along open or quasi-open magnetic filed lines and they can induce Langmuir waves which will the cause the radio emission at plasma frequency and/or its harmonic.**

   **Answer:** We thank the referee. The paragraph is modified, refer to Page 2, lines 32 -36.

3. **Page 2, line 26: The authors write that type III bursts duration is 1-3 s. However, as duration of the type IIIs depends on the observing frequency it is necessary to specify the frequency range together with the durations.**

   **Answer:** The statement is formulated as follows: Type III radio bursts drift from $\sim$ 200 MHz to 30 MHz in less than 10 s and reach 30 kHz in about 1 hour. Page 2, lines 37 -38.

4. **Page 2, line 27: The authors say: "The impulsive flares in X-ray and/or $H_\alpha$ frequencies..." It should be x-ray and $H_\alpha$ wavelengths.**

   **Answer:** We thank the referee. The error is corrected. Page 2, line 39

5. **Page 2, line 31: The authors already used abbreviation "CME" so no need to use the full name.**

   **Answer:** The abbreviation is maintained.

6. **Page 2, line 36: The authors should also mention very important and recently refurbished Nançay observations.**

   **Answer:** We have added the recommended observatory. Page 3, lines 76 - 77

7. **Page 2 footnoote 3. The link for the "callistoQuicklooks" does not work.**

   **Answer:** We have provided a correct link, `http://soleil.i4ds.ch/solarradio/callistoQuicklooks/`. Page 4

8. **Page 3, line 58: What does mean 'through a channel of 80.9 MHz to 45 MHz?**

   **Answer:** The statement is rephrased as follows. .... radio bursts were identified within 45 - 80.9 MHz frequency range. Page 4, line 115

9. **Table 1: The authors do not define when they consider the radio burst to be "associated with flares and/or CMEs". What is the time window in which two phenomena are considered to be related?**

   **Answer:** Page 6, lines 126 – 129

   (a) To confirm the CME -SRBs association, we checked their time coincidence (for this study, CMEs that occurred within one hour of the onset of the radio bursts were considered). we used white-light coronagraph data from the Large Angle and Spectrometic Coronagraph (LASCO C2) onboard the Solar and Heliospheric Observatory (SOHO) and the Sun-Earth Connection Coronal and Heliospheric Investigation (SECCHI) onboard the Solar Terrestrial Relationship Observatory (STEREO) Ahead (A) and Behind (B) spacecrafts. For some cases where the identification was not clear, the CMEs were checked manually around the time of the moving burst events. For the events where multiple CMEs occurred at the same times, we checked the source region of the movingbursts from CALLISTO images to associate them with the correct CME.

   (b) Regarding flare association, the onset, peak and end time of the flare were checked from the X-ray flare plot from the Geostationary Operational Environmental Satellite (GOES). We confirmed the association by checking their time coincidance, most likely when the radio burst is present between the onset and end times of the associated flare, we treat it as flare associated .

10. **Page 4, line 75: The authors state wrongly: "type IV bursts may be due to small-scale feature events present in the solar corona. First, the type IV bursts are generally associated with CMEs, so they cannot be put in the same category as type III bursts that are not flare associated. Second what are 'small-scale feature events'? Do the authors refer to nanoflares, small scale reconnection events or something else?**

   **Answer:** We thank the referee for this comment. Type III radio bursts and type IVs are separate, no way toput them in the same category. The two lines are modified as follows.The remaining non-flare associated type III radio bursts may be due to small-scale reconnection events present in the solar corona. Page 7, lines 145 – 146

11. **Page 4, line 78: The authors write incorrect, type II bursts and not type IV radio bursts are triggered by CME-driven shocks. T he type IV emission is considered to be associated with CMEs, but not CME-driven shocks.**

   **Answer:** The statement is rephrased as follows. The majority ( 82%) of type IV radio bursts are associated with CMEs which is consistent with the results obtained by Kumari et al. (2021) who analyzed a set of 446 type IV radio bursts and found that

81% of them are temporally and spatially correlated with CMEs., Page 7, lines 149 – 152.

12. **Page 4, line 79: On figure 2 there is absolutely no information that would indicate that studied event was geoeffective. So, the following statement has no ground: "the one of August 22, 2015 is chosen based on its geoeffectiveness as displayed in Figure 2."**

    **Answer:**

    (a) The statement is rephrased as follows: Among type II bursts detected by the instrument, a case study of the 5 November 2014 event is taken. Figure 3 (a) shows a type II radio spectrum recorded on 5 November 2014 by CALLISTO spectrometer based at Rwanda. Page 7, lines 153 - 154.

    (b) Initial Figure 2 was dropped because it is fully described in another paper, see http://doi.org/10.1016/j.asr.2021.06.029.

13. **Figure 2 (Page 7): Panel c) does not show the CME itself but it shows the associated EUV wave, one of the on disc signatures of CME. Considering blue circle as a CME-driven shock is very provisional consideration, and needs to be better justified and with some references.**

    **Answer:** This description was removed since Figure 2 was dropped.

14. **Page 4, line 81: What means "The event has a band split fundamental structure with the corresponding frequencies ranging between 46-56 MHz and 46-75 MHz,"? I believe it should be written something like: the fundamental band of the type II burst shows the band split. It should be also explained what is band split, and what are the listed frequencies? References need to be added describing the band split, eg. Vrsnak et al., 2001 and some recent work like Mahrous et al., 2018.**

    **Answer:** We thank the referee for the useful comment and suggestion. The references are added in the literature. The same feature is discussed for a 5 November 2014 type radio burst in the revised manuscript. Page 7–8, lines 154 – 158.

15. **Page 4, line84: The authors state that the CME occurred at 07:12 UT. Where did he occurred, in the EUV images or in coronagraph images, and which one SOHO/LASCO C2? All this needs to be specified.**

    **Answer:** The sentence is revised and rephrased for a 5 November 2014. Page 8, lines 162 – 163

16. **Page 4, line 85-86: I am wondering how the CME with the speed of about 640 km/s arrived from the Sun to Earth in less then 24 hours?! This is surly not possible. The authors should check the association of the solar event and the one observed in situ. The disturbance propagating with the speed of about 700 km/s will need about 2.5 days to come to Earth. Further, the way shock height is estimated is not scientifically justified.**

**Answer:** We thank the referee for this comment. Depending on the speed, a typical CME reaches the Earth between 1 – 5 days. But the statement was dropped becaused linked event was removed.

17. **Page 5, line 88: What does mean that flare occurred at 06:49 UT? Was that the flare start of the maximum? All this needs to be specified.**

    **Answer:** The statement is modified for 5 November 2014.

18. **Page 5, line 89-90: The simultaneous occurrence of the flare and CME is not the point which makes difficult understanding which one of them is generating coronal shock. It is the simultaneous flare impulsive phase and the CME acceleration phase that does not allow us to clearly understand if the shock is flare-generated of CME-driven.**

    **Answer:** We thank the referee for this important comment. The sentence is revised.

19. **Figure 3 (page 8): In the presented images no clear jets are observed. What the authors call bright 'nods' are the brightenings observed when the closed loop system moves. Usually, jets are observed propagating along the open filed lines, and I see nothing like this in the presented figure. And in particular, that should be the case when jets are associated with type III radio bursts – signatures of fast electron beam propagating along open field lines. We can observe open field lines in the figure but none of them shows brightening indicating jet propagation.**

    **Answer:** Figure 3 is shifted to Figure 4. It aims to show the potential site where non-flare type III radio bursts are believed to originate. We have looked at the Sun and taken portions of the Sun containing the region of interest. We have increased the size of the images and modified the figure caption. Page 19.

20. **Page 5, line 102-103: This sentence has no sense. Where the authors see propagating waves?**

    **Answer:** We thank the referee. we have removed the sentence.

21. **Page 5, line 104-105: This sentence is not justified. In order to associate type III bursts with CME at least the source positions of type III bursts need to be checked and compared with the source region of CMEs. The authors should study literature a bit better. Namely, numerous type III bursts can appear during the so called type III storms that can last for days. And, it is generally considered that they are associated with the complex active regions observed on the visible side of the solar disc.**

    **Answer:** We thank the referee. We used the CME catalog and images at `https://helioviewer.org` to confirm the association between type III bursts and CMEs. Since we did not check their source position, we have dropped the statement.

22. **Page 5, line 113: What does this sentence mean?**

    **Answer:** We thank the referee. The sentence was removed as it has no meaning.

23. **Page 5, line 114: What is 'backbone of the CME-driven shocks'? Why the authors think type IV bursts are 'poorly associated with flares? This statement needs to be justified and references provided.**

    **Answer:** The statement is rephrased and modified. Table 2 lists all type IV radio bursts observed by the instrument and their association with the solar phenomena. From this table, it can be seen that type IV bursts are highly associated with CMEs which indicates that, their presence can be used to map the location of trapped electrons and studying the CME kinematics during the phases of eruption processes (Kumari et al., 2021). Page 8 – 9, lines 189 – 192.

24. **Page 6, line 117 – 118: Figure 6 does not show any type IV radio bursts, but it shows somehow distorted images of CMEs in the SOHO/LASCO C2 filed of view. The running difference images of SDO, 171 A are also very bad. It is not completely clear where is the solar limb and the Sun seems distorted.**

    **Answer:**

    (a) Figure 6 aims to show the images of CMEs and their corresponding potential site on the solar surface for the dates where non-flare associated type IV radio bursts were detected. The figure does not indicate any type IV radio burst at all.

    (b) The figure shifted to Figure 7. Page 21.

    (c) We now modify the figure as. Figure 7 presents the magnetic loops embedded in the solar corona routinely the site of trapped electrons responsible for Type IV emissions. The figure displays a sample of six AIA/SDO images in a 171 Å band filter corresponding to the dates of type IV radio bursts in January 2015. Page 9, lines 195 – 197.

25. **Page 6, line 118-119: How is this visible?**

    **Answer:** We have dropped the statement as an effect of Figure 6 reformulation.

26. **Page 6, line 122-124: The authors first state the some type IV radio burst may be not associated with flares and CMEs. And in the following sentence they state that this kind of type IV bursts coincide with the decaying phase of flares or post eruption loops. These two sentences are in contradiction.**

    **Answer:** The sentences are modified. Recent results showed that type IV radio bursts coincide with the decaying phase of flares and/or triggered by post-eruption loops (e.g., Morosan et al., 2019; Kumari et al., 2021). Moreover, these bursts may lack both association with flares and CME eruptions (Table 2). Pages 9 – 10, lines 197 – 199.

27. **Page 6, line 124: This is very general sentence, not related at all to presented study. It cannot be considered as conclusion.**

    **Answer:** We thank the referee for this comment. The statement is dropped.

28. **Page 7, line: By e-CALLISTO we can be continuously monitor radio emission and not Space Weather.**

    **Answer:** We thank the referee. The statement is corrected.

29. **Page 9, line 141: Which radio bursts may be used as a precursor for space weather diagnostics? And how? The authors did not really provide convincing evidence, actually any evidence how we can use radio observations for space weather diagnostics.**

    **Answer:**

    (a) Observed radio bursts (type II, III and type IV) may be used as a precursor for space weather diagnostics since they are often associated with space weather drivers such as CMEs and SFs. Solar radio bursts can be detected at the ground level, hence serving as the advance warning of incoming associated solar transient events. Therefore, they provide insights to predict/forecast the incoming space weather hazards. HF communications and the ionosphere are disturbed by the X-ray and EUV wavelengths along with the solar energetic particles that reach Earth within few hours and they are signatures of solar flares and CMEs.

    (b) We have described this part in section 4.2 of the revised manuscript with appropriate examples.